

# Novel monoclonal antibodies for immunodetection of AmpC β-lactamases

Karolina Bielskė[1], Martynas Simanavičius[1], Julie Nuttens[2],
Julija Armalytė[3], Justas Dapkūnas[1], Lukas Valančauskas[1] and
Aurelija Žvirblienė[1]

[1] Institute of Biotechnology, Life Sciences Center, Vilnius University, Vilnius, Lithuania
[2] ArcDia International Ltd, Turku, Finland
[3] Institute of Biosciences, Life Sciences Center, Vilnius University, Vilnius, Lithuania

## ABSTRACT

**Background:** Accurate and easy-to-perform assays for the detection of antibiotic-resistant bacterial isolates producing AmpC β-lactamases are epidemiologically relevant, leading to more effective use of antibiotics and a comprehensive understanding of β-lactamase prevalence. We describe novel monoclonal antibodies (MAbs) against CMY β-lactamases and their application in immunoassays for the detection of CMY-producing bacterial isolates.

**Methods:** Recombinant CMY-34 was expressed in *Escherichia coli* and used as an immunogen for MAb generation by hybridoma technology. Selected CMY-34-specific MAbs were comprehensively characterized by various immunoassays, computational analysis, and sequencing of their variable domains. To prove MAb reactivity with CMY β-lactamases, the antibodies were tested with CMY-producing bacterial isolates. For this purpose, the MAbs were applied in sandwich-type assays, such as sandwich enzyme-linked immunosorbent assay (ELISA), lateral flow immunoassay (LFIA), and two-photon excitation (TPX) assay for immunodetection of CMY enzymes.

**Results:** Two high-affinity MAbs raised against recombinant CMY-34 were characterized in detail. Both MAbs recognized CMY-34 β-lactamase in the *Citrobacter portucalensis* isolate. The analysis of MAb epitopes revealed their sequence homology among the members of the CMY family, suggesting their potential broad reactivity. Comprehensively characterized MAbs were successfully applied in sandwich ELISA and two rapid immunoassay formats that were tested with CMY-positive bacterial isolates. MAb-based immunoassays detected all analyzed CMY-positive isolates producing CMY-2, CMY-4, CMY-6, CMY-16, and CMY-34 β-lactamases.

**Conclusion:** Novel MAbs raised against CMY-34 recognize common epitopes of CMY β-lactamases and can be applied for immunodetection of CMY β-lactamases in bacterial isolates.

Corresponding author
Karolina Bielskė,
karolina.bielske@gmc.vu.lt

## INTRODUCTION

The increasing prevalence of antibiotic resistant bacteria poses a critical risk to human health globally (*Barriere, 2015*; *Hussein, AL-Kubaisy & Al-Ouqaili, 2024*). The emergence of AmpC β-lactamases (further abbreviated as AmpCs) in Gram-negative bacteria is widely identified in health care settings, and AmpCs are increasingly being detected in livestock, wild and companion animals (*Jacoby, 2009*; *Ewers et al., 2012*; *Madec et al., 2017*). AmpCs are clinically important cephalosporinases, which are also classified as class C β-lactamases (*Ambler, 1980*; *Bush & Jacoby, 2010*). The production of these enzymes confers high-level resistance to cephalosporins and penicillin-β-lactamase inhibitor combinations (*Jacoby, 2009*). There are two resistance mechanisms of AmpCs identified: plasmid-mediated resistance, which is increasingly being detected in *Escherichia coli*, *Klebsiella* spp., *Salmonella* spp. and other species, and chromosomally encoded inducible or noninducible resistance identified in *Citrobacter* spp., *Pseudomonas* spp., *Enterobacter* spp., *Acinetobacter* spp., *E. coli*, *Shigella* spp. and other species (*Jacoby, 2009*; *Drawz & Bonomo, 2010*; *Pfeifer, Cullik & Witte, 2010*). The degree of resistance depends on the expression level of AmpCs and the presence of other resistance mechanisms (*Jacoby, 2009*).

The CMY-2-like enzymes are currently identified as one of the most common and widely disseminated AmpCs (*Jacoby, 2009*; *European Committee on Antimicrobial Susceptibility Testing, 2017*). CMY family of β-lactamases encoding genes ($bla_{CMY}$) have been identified on both the chromosomes and plasmids of Gram-negative bacteria, such as *E. coli*, *Klebsiella* spp., *Salmonella* spp., *etc.* (*Jacoby, 2009*). Currently, more than 180 allelic variants of $bla_{CMY}$ are identified and provided in the Beta-Lactamase DataBase (BLDB) (http://www.bldb.eu/, last update of the database: January 22, 2025) (*Naas et al., 2017*). CMY-2 β-lactamase remains the most prevalent and clinically significant AmpC enzyme, however, other CMY variants, such as CMY-34 β-lactamase, have emerged (*Jacoby, 2009*; *Naas et al., 2017*; *Seo et al., 2023*; *Rahman et al., 2025*). Several cases of $bla_{CMY-34}$ detection were reported. In 2007, CMY-34 coding gene was identified by sequencing genomic DNA of multidrug-resistant *Citrobacter freundii* (*Zhu, Xu & Xu, 2007*). After a few years, $bla_{CMY-34}$ was detected in *Citrobacter* spp. isolated from the feces of army recruits and fecal *E. coli* isolated from horses subjected to antimicrobial treatment in Denmark (*Hammerum et al., 2011*; *Damborg et al., 2012*). Later, CMY-34 coding gene was identified in *C. freundii*, which was isolated from urban aquatic environment (*Manageiro et al., 2014*). Despite the identified and reported cases of CMY-34 detection, this β-lactamase remains one of the barely investigated variants within the CMY family.

AmpC producing Gram-negative bacteria pose a major challenge in the treatment of infectious diseases (*Rodríguez-Baño et al., 2018*). The selection of antibiotics for treatment and monitoring of bacterial susceptibility to antibiotics differ across clinical settings. Variations in the prevalence of infectious diseases lead to differences in antibiotic resistance mechanisms across geographical regions. Therefore, the epidemiological surveillance plays a crucial role in the assessment and tracking of resistance profiles in bacteria enabling effective interventions. Obtained epidemiological data is beneficial for

making treatment recommendations and implementing appropriate guidelines for effective use of antibiotics at local, national or global levels (*Altorf-van der Kuil et al., 2017*; *Diallo et al., 2020*). Therefore, accurate and easy-to-perform diagnostic assays for detection of AmpCs producing bacteria are epidemiologically relevant, leading to more effective use of antibiotics.

The European Committee on Antimicrobial Susceptibility Testing (EUCAST) recommends several phenotypic methods for AmpCs detection in bacterial isolates. For instance, detection of resistance to cefoxitin combined with resistance to ceftazidime and/ or cefotaxime is proposed as phenotypic criteria for evaluation of AmpCs producers (*European Committee on Antimicrobial Susceptibility Testing, 2017*). For additional confirmation of AmpC production, the phenotypic techniques with cloxacillin or boronic acid derivatives can be applied (*European Committee on Antimicrobial Susceptibility Testing, 2017*). Commercial tests are also available for AmpC detection, such as AmpC Detection Disc Set (Mast, Bootle, UK), AmpC gradient test (bioMerieux, France) and tablets (Rosco, Stamford, CT, USA) containing cefotaxime-cloxacillin and ceftazidime-cloxacillin (*Ingram et al., 2011*; *Halstead, Vanstone & Balakrishnan, 2012*; *Hansen et al., 2012*). As alternative, the presence of AmpCs can be confirmed using polymerase chain reaction (PCR)-based or DNA microarray-based methods (*Pérez-Pérez & Hanson, 2002*; *Cuzon et al., 2012*; *European Committee on Antimicrobial Susceptibility Testing, 2017*).

Several assays using nucleic acid-based, biochemical and antibody-based approaches for detection of CMY enzymes have been described. Recombinase polymerase amplification method for CMY-2 variant, real-time PCR and DNA microarray for CMY-type detection have been reported (*Cuzon et al., 2012*; *Hoj et al., 2021*; *Ertl et al., 2023*). Direct MALDI-TOF mass spectrometry was successfully applied for CMY-2 testing (*Espinosa et al., 2018*). Enzyme-linked immunosorbent assay (ELISA) applying CMY-2-specific nanobodies and rabbit polyclonal IgG for detection and quantification of CMY-2 in bacterial isolates have been developed (*Hujer et al., 2002*; *Frédéric et al., 2022*). Moreover, a neutralization test for CMY-type enzymes utilizing rabbit antiserum against CMY-2 has been described (*Attia, Fatah & El-mowalid, 2017*).

Recently, a novel bioaffinity platform based on proprietary two-photon excitation microfluorometric technology (TPX) has been described and successfully applied for *in vitro* diagnostics of respiratory and gastrointestinal infections (*Soini et al., 2002*; *Waris et al., 2002*; *Koskinen et al., 2018*, *2021*). This technology is characterized by separation-free fluorescence detection using an analyzer that monitors reaction kinetics in real time. The TPX platform allows antimicrobial susceptibility testing directly from polymicrobial clinical samples (*Koskinen et al., 2008*).

Although a sandwich ELISA has been applied for CMY-2 detection, no such assay or monoclonal antibodies (MAbs) capable of detecting other CMY β-lactamases have been described yet. Moreover, rapid tests such as TPX assay or lateral flow immunoassay (LFIA) for immunodetection of CMY-type β-lactamases in bacterial samples are not available yet. In this study, we describe novel broadly reactive MAbs raised against one of the allelic

variants of CMY β-lactamases—CMY-34—and their application in sandwich ELISA, LFIA and TPX assays for immunodetection of CMY β-lactamases in bacterial isolates.

# MATERIALS AND METHODS

## Cloning and expression of recombinant CMY-34

A synthetic and codon-optimized gene (Invitrogen, Waltham, MA, USA) encoding CMY-34 (GenBank accession no. EF394370.1) with BamHI restriction site at 5′-end and XhoI site at 3′-end was cloned into the respectively hydrolyzed pET28a(+) vector (Novagen, Irene, South Africa), fusing the gene to His-tag coding sequence at the 5′-end of the gene. The construct was verified by Sanger sequencing (GENEWIZ), and the plasmid was subsequently transformed into *E. coli* Tuner (DE3) (Novagen, Irene, South Africa). For recombinant CMY-34 (rCMY-34) expression, an overnight culture of *E. coli* Tuner (DE3) was diluted (dilution factor 1:100) and cultivated in 1 L flasks with Luria-Bertani (LB) medium supplemented with 30 μg/mL kanamycin by shaking (220 rpm) at 37 °C until the optical density (OD) at 600 nm reached 0.6. The expression was induced with 0.2 mM isopropyl β-D-1-thiogalactopyranoside (IPTG) for 4 h at 25 °C by shaking. Then, the culture was pelleted at 3000× *g* for 15 min at 4 °C, washed with buffer (0.2 M NaCl, 20 mM Tris-HCl, pH 7.4), centrifuged again and suspended in the purification buffer (0.15 M NaCl, 20 mM imidazole, 10 mM β-mercaptoethanol, 20 mM Tris-HCl, pH 7.4) supplemented with 1 mM phenylmethylsulfonyl fluoride (PMSF). The suspension was stored at −70 °C until rCMY-34 purification.

## Purification of rCMY-34

The frozen culture suspension was thawed on ice and disrupted by sonication. The supernatant was cleared by centrifugation at 15,000× *g* for 20 min at 4 °C, filtered through 0.45 μM polyvinylidene difluoride (PVDF) filter and diluted with purification buffer. Purification was performed using an ÄKTA™ start chromatography system (29022094; Cytiva, Marlborough, MA, USA) and 1 mL HisTrap™ HP column (29051021; Cytiva, Marlborough, MA, USA) according to the manufacturer's recommendations. The elution step was performed with the imidazole gradient (80–500 mM). The purity of rCMY-34 was analyzed by sodium dodecyl sulfate–polyacrylamide gel electrophoresis (SDS-PAGE) under reducing conditions, selected fractions were pooled, and the buffer was exchanged into storage solution (2 mM dithiothreitol (DTT), 0.2 M NaCl, 20 mM Tris–HCl, pH 7.4).

## Generation of monoclonal antibodies against CMY-34

For generation of CMY-34 specific MAbs, three 6–8 week old female BALB/c mice were immunized subcutaneously with 50 μg of rCMY-34 three times as described previously (*Zvirbliene et al., 2010*). The sample size was determined based on standard laboratory practices and the hybridoma technology protocol following the guidelines specified in DIRECTIVE 2010/63/EU. No randomization, control group, or blinding procedures were implemented, and no statistical methods were applied for MAb generation. During immunization, the blood samples were collected by tail bleeding, and the titer of antigen-specific IgG in the blood was evaluated by indirect ELISA. Three days before

hybridization, the mouse that generated the highest titer of antigen-specific IgG was boosted subcutaneously with 50 μg of rCMY-34 in phosphate–buffered saline (PBS). Animal care and experimental protocols were adhered to the ARRIVE and FELASA guidelines in accordance with European and Lithuanian legislations. ARRIVE study plan is available at MIDAS repository (http://dx.doi.org/10.18279/MIDAS.258798). All immunized mice were sacrificed without anesthesia by cervical dislocation with the requirements specified in ANNEX IV of DIRECTIVE 2010/63/EU. The killing of mice was completed by confirmation of the onset of *rigor mortis*. No live mice were left at the end of the experiment. The hybridization was performed as described by *Köhler & Milstein (1975)*. Briefly, the spleen cells of immunized mouse were isolated and fused with mouse myeloma Sp2/0 (ATCC, CRL-1581) using polyethylene glycol PEG-4000 (P7306; Sigma-Aldrich, Burlington, MA, USA), and then cultivated in Dulbecco's modified Eagle's medium (DMEM) supplemented with 15% fetal bovine serum (FBS) and selection reagent containing hypoxanthine, aminopterin and thymidine (HAT, Sigma-Aldrich, Burlington, MA, USA, H0262). Then, secretion of antigen specific MAbs in the hybridoma cell culture supernatant was tested by indirect ELISA. Selected hybridoma clones were cloned using limiting dilution method, additionally tested by indirect ELISA, cryopreserved and cultivated for MAb purification.

Mice used for hybridization were obtained from Vilnius University, Life Sciences Center, Institute of Biochemistry (Vilnius, Lithuania), which has State Food and Veterinary Agency (Vilnius, Lithuania) permission to breed and use experimental animals for scientific purposes (vet. approval no. LT 59–13–001, LT 60–13–001, LT 61–13–004). Ethical approval to use BALB/c mice for experiments was granted by State Food and Veterinary Agency (Vilnius, Lithuania), permission no. G2–117, issued 11 June 2019. Mice care and experimental procedures were carried out by trained personnel in accordance with Directive 2010/63/EU. Mice were monitored daily and kept in controlled conditions (12 h light-dark cycle, temperature of $22 \pm 1\,^{\circ}\mathrm{C}$, humidity $55 \pm 3\%$, cardboard enrichment) and had constant access to standard rodent food and water *ad libitum*. No analgesia or anesthesia was administered during any procedures following the hybridoma technology protocol and bioethics approval.

MAbs were purified from hybridoma cell culture supernatants by affinity chromatography as described previously (*Sližienė et al., 2022*).

## Enzyme-linked immunosorbent assay (ELISA)

Indirect ELISA was used for testing of blood samples of immunized mice, selection of antigen-specific MAb producing hybridoma cells, cross-reactivity testing and determination of apparent dissociation constants ($K_d$). A total of 96-well plates (10547781; Thermo Scientific, Waltham, MA, USA) were coated with 50 μL/well of antigen at 3–5 μg/mL concentration in coating buffer (50 mM sodium carbonate, pH 9.5) for 16 h at 4 °C, and blocked with 250 μL/well of 2% bovine serum albumin (BSA) (w/v) in PBS for 1 h at room temperature (RT). After blocking, the wells were incubated with mouse blood samples (dilution ranging from 1:300 to 1:656,100 in phosphate–buffered saline containing 0.1% Tween-20 (v/v) (PBST)) or with undiluted hybridoma cell culture supernatant.

For $K_d$ determination, MAbs in the concentration range of 32–0.005 nM in PBST were tested. After incubation, plates were washed five times with PBST, followed by incubation with 50 µL/well of secondary horseradish peroxidase (HRP)-conjugated goat anti-mouse IgG antibodies (1721011; Bio-Rad, Hercules, CA, USA) (dilution factor 1:5,000 in PBST) for 1 h at RT. Then, the plates were washed six times with PBST and incubated with 50 µL/well of 3,30,5,50-tetramethylbenzidine (TMB, Clinical Science Products, 01016-1-1000). The reaction was stopped with 25 µL/well of 1 N $H_2SO_4$ solution. The OD was measured with microplate spectrophotometer and calculated as difference between measured OD values at 450 nm and reference wavelength at 620 nm.

The $K_d$ values of purified MAbs were determined from their titration curves. The determined value indicates MAb concentration (nM) at which the OD decreased by 50%.

Sandwich ELISA was used for development of CMY detection system. For the assay, 96-well plates (10547781; Thermo Scientific, Waltham, MA, USA) were coated with capture MAb (100 µL/well) at concentration of 2.5 µg/mL in coating buffer for 16 h at 4 °C. The wells were blocked with 250 µL/well of ROTI® Block (A151.1; Carl Roth, Karlsruhe, Germany) for 1 h at RT. Then, the plates were incubated with 100 µL/well of serially diluted rCMY-34 as a standard in the range of 90–0.123 ng/mL or bacterial lysates with known total protein concentration in the range of 5,000–6.68 ng/mL in PBST for 1 h at RT. *E. coli* BL21 (Novagen, Irene, South Africa) lysate was tested as β-lactamase non-producing control. Subsequently, the wells were washed five times with PBST and incubated with 100 µL/well of HRP-labeled detection MAb for 1 h at RT. After incubation, the plates were washed six times with PBST, and 100 µL of TMB solution was added to each well to stop the reaction as described above.

Direct ELISA was used for testing of HRP-conjugated MAbs. For the assay, the antigen coating, blocking, washing and other procedures were performed as described above, in the protocol of indirect ELISA. The HRP-conjugated MAbs were serially diluted in the range of 1:100–1:72,900 with PBST (50 µL/well).

Competitive ELISA was used for selection of non-competing pairs of MAbs, suitable for sandwich-type immunoassays. For the method, antigen coating, blocking, washing and other procedures were carried out as described above, in the protocol of indirect ELISA. After blocking step, the plates were incubated with 50 µL/well of MAbs at concentration of 10 µg/mL for 1 h at RT. Then, HRP-conjugated MAbs (50 µL/well) were added, and the plates were incubated for 1 h at RT.

The isotypes of MAbs were determined using Mouse Immunoglobulin Isotyping ELISA Kit (550487; BD Biosciences, Franklin Lakes, NJ, USA) according to the manufacturer's recommendations.

MAbs were conjugated with HRP as described previously (*Stravinskiene et al., 2019*).

The epitope of MAb 9D2 against CMY-34 was identified using a set of N-biotinylated synthetic peptides (GenScript) spanning a 22–49 amino acid (aa) region of CMY-34 (GenBank accession no. EF394370.1) and containing a linker sequence (-SGSG-) (Table 1). For testing, 96-well plates (10547781; Thermo Scientific, Waltham, MA, USA) were coated with 50 µL/well of Pierce™ avidin (21128; Thermo Scientific, Waltham, MA, USA) at concentration of 5 µg/mL in deionized water for 16 h at 4 °C, and blocked with 250 µL/well

**Table 1 Description of synthetic peptides used for epitope mapping.**

| Synthetic peptide | Region of CMY-34, aa | aa sequence[a] |
|---|---|---|
| P0 | 22–49 | **SGSG**AKTEQQIADIVNRTITPLMQEQAIPGMA |
| P1 | 25–49 | **SGSG**EQQIADIVNRTITPLMQEQAIPGMA |
| P2 | 28–49 | **SGSG**IADIVNRTITPLMQEQAIPGMA |
| P3 | 31–49 | **SGSG**IVNRTITPLMQEQAIPGMA |
| P4 | 34–49 | **SGSG**RTITPLMQEQAIPGMA |
| P5 | 37–49 | **SGSG**TPLMQEQAIPGMA |
| P6 | 40–49 | **SGSG**MQEQAIPGMA |
| P7 | 43–49 | **SGSG**QAIPGMA |
| P8 | 46–49 | **SGSG**PGMA |
| P9 | 22–46 | **SGSG**AKTEQQIADIVNRTITPLMQEQAIP |
| P10 | 22–43 | **SGSG**AKTEQQIADIVNRTITPLMQEQ |
| P11 | 22–40 | **SGSG**AKTEQQIADIVNRTITPLM |
| P12 | 22–37 | **SGSG**AKTEQQIADIVNRTIT |
| P13 | 22–34 | **SGSG**AKTEQQIADIVNR |
| P14 | 22–31 | **SGSG**AKTEQQIADI |
| P15 | 22–28 | **SGSG**AKTEQQI |
| P16 | 22–25 | **SGSG**AKTE |
| Full-length CMY-34[b] | 1–381 | |

**Notes:**
[a] Peptides contain N-terminal biotin; bolded aa correspond to linker sequence (-SGSG-).
[b] According to CMY-34 protein sequence: GenBank accession no. ABN51006.1.

of 2% BSA (w/v) in PBS for 1 h at RT. After blocking step, the plates were washed and incubated with 100 μL/well of synthetic peptides at 10 μg/mL concentration in PBST for 1 h at RT. Then, the plates were washed and incubated with 100 μL/well of MAb 9D2 at 5 μg/mL concentration in PBST for 1 h at RT. The following steps of incubation with HRP-conjugated goat anti-mouse IgG antibodies and TMB were carried out as described above.

## Western blot analysis (WB)

The reactivity of MAbs with β-lactamases was evaluated by western blot analysis (WB) as described previously (*Kucinskaite-Kodze et al., 2020*). Briefly, recombinant proteins (0.5–1 μg/line) or bacterial lysates (7–10 μg/line) were mixed with Pierce™ Lane Marker Reducing Sample Buffer (39000; Thermo Scientific, Waltham, MA, USA), boiled and fractionated by SDS-PAGE using 12% or 15% polyacrylamide gels. *E. coli* BL21 lysate was tested as β-lactamase non-producing control. After protein transfer to 0.45 μm PVDF membrane, the blocking of the membrane with 2% milk powder (w/v) in PBS was performed for 1 h at RT, followed by incubation with MAbs at 3 μg/mL concentration for 1 h at RT. Then, the membrane was washed with PBST and incubated with secondary goat anti-mouse IgG-HRP antibodies (1721011; Bio-Rad, Hercules, CA, USA) (dilution rate 1:4,000 with 2% milk powder in PBST) for 1 h at RT. The membrane was washed with

**Table 2 Description of PCR primers and CMY-34 fragments used for epitope mapping.**

| CMY-34 fragment | Region of gene, bp[b] | Region of protein, aa[b] | PCR primer sequence (5′–3′)[a] |
|---|---|---|---|
| CMY-34_1 | 1–405 | 1–135 | ACAGGATCCATGATGAAAAAAAGCCTGTG<br>ACACTCGAG**TTA**ATCACCACCTAAAACACC |
| CMY-34_2 | 61–405 | 22–135 | ACAGGATCCGCAGCAAAAACCGAACAG<br>ACACTCGAG**TTA**ATCACCACCTAAAACACC |
| CMY-34_3 | 148–405 | 50–135 | ACAGGATCCGTTGCAGTTATTTATCAGGGT<br>ACACTCGAG**TTA**ATCACCACCTAAAACACC |
| CMY-34_4 | 292–876 | 98–292 | ACAGGATCCATTGCCCGTGGTGAAATC<br>ACACTCGAG**TTA**TTCCCAACCTAAACCTTG |
| CMY-34_5 | 523–876 | 175–292 | ACAGGATCCATTGGTCTGTTTGGTGC<br>ACACTCGAG**TTA**TTCCCAACCTAAACCTTG |
| CMY-34_6 | 523–1,143 | 175–381 | ACAGGATCCATTGGTCTGTTTGGTGC<br>ACACTCGAG**TTA**CTGCAGTTTTTCCAGAATAC |
| Full-length CMY-34[b] | 1–1,146 | 1–381 | |

**Notes:**
[a] Underlined nucleotides correspond to restriction sites of BamHI and XhoI; bolded nucleotides correspond to STOP codon.
[b] According to CMY-34 coding DNA and protein sequences: GenBank accession no. EF394370.1 and ABN51006.1, respectively.

PBST and NeA-Blue Precip reagent (Clinical Science Products, 01283-1-200) was used for detection.

## Immunoprecipitation (IP)

The reactivity of MAbs with β-lactamases was tested by IP. For analysis, 125 µL of rProtein G Sepharose™ Fast Flow resin (17127905; Cytiva, Marlborough, MA, USA) was calibrated by washing the resin four times with 0.1 M Tris-HCl (pH 8) buffer and centrifuging at 12,000× $g$ for 30 s. After calibration, the resin was blocked with ROTI® Block (A151.1; Carl Roth, Karlsruhe, Germany) for 1 h at RT with rotation. Then, the resin was washed with 0.1 M Tris-HCl (pH 8) buffer, mixed with 25 µg of MAb and incubated for 1 h at RT with rotation. Formed complexes were washed four times with PBST, divided into five separate tubes and incubated with bacterial lysates with known total protein concentration (40 µg/tube) or rCMY-34 (5 µg/tube) for 1.5 h at RT with rotation. After incubation, the complexes were washed four times with PBST, and the elution step was performed with 0.1 M glycine (pH 3) buffer. The eluted fractions were mixed with Pierce™ Lane Marker Reducing Sample Buffer (39000; Thermo Scientific, Waltham, MA, USA), boiled and analyzed by SDS-PAGE and WB. For WB, an HRP-conjugated MAb 9D2 against CMY-34 (dilution factor 1:200 with 2% milk powder (w/v) in PBST) was used for detection.

## Epitope mapping

Epitope mapping of newly generated MAbs was performed using truncated overlapping CMY-34 fragments. In the first step, the potential linear B-cell epitopes of CMY-34 protein were predicted using the BepiPred-2.02 tool (https://services.healthtech.dtu.dk/services/BepiPred-2.0/) (*Jespersen et al., 2017*), which divided the antigen into six overlapping fragments (Table 2).

The DNA sequences of CMY-34 fragments were amplified by PCR using primers with additional BamHI and XhoI restriction sites (Table 2) and a synthetic CMY-34 gene

**Table 3 Description of PCR primers used for amplification of MAb variable regions.**

| PCR primer | PCR primer sequence (5′–3′)[a] | Amplified chain | Reference |
|---|---|---|---|
| IgG1 | TTAATAGACAGATGGGGGTGTCGTTTTGGC | Heavy | *Wang et al. (2000)* |
| MH1 | CATATGSARGTNMAGCTGSAGSAGTC | | |
| Kc | TTAGGATACAGTTGGTGCAGCATC | Light | |
| Mk | CATATGGAYATTGTGMTSACMCARWCTMCA | | |
| VH1FOR | TGAGGAGACGGTGACCGTGGTCCCTTGGCCCCAG | Heavy | *Orlandi et al. (1989)* |
| VH1BACK | AGGTSMARCTGCAGSAGTCWGG | | |
| VK2FOR | GTTATTTGATCTCCAGCTTGGTCCC | Light | |
| VK1BACK | AGGTSMARCTGCAGSAGTCWGG | | |

**Note:**
[a] Modified sequence of the primer is underlined.

(GenBank accession no. EF394370.1) as a template. The amplified products were hydrolyzed with restriction nucleases, cloned into the respectively hydrolyzed pET28a(+) vector and verified by Sanger sequencing (GENEWIZ). For expression of His-tag-fused fragments, an overnight culture of *E. coli* Tuner (DE3) was diluted (dilution factor 1:100) and cultivated in 5 mL of LB supplemented with 30 μg/mL kanamycin by shaking (220 rpm) at 37 °C until the OD at 600 nm reached 0.8–1. Protein expression was induced with 1 mM IPTG for 2.5 h at 37 °C by shaking. The cells were sedimented by centrifugation at 3,000× *g* for 5 min, mixed with 1% sodium dodecyl sulfate (SDS) (w/v) in PBS and boiled. Then, Pierce™ Lane Marker Reducing Sample Buffer (39000; Thermo Scientific, Waltham, MA, USA) was added, and samples were boiled repeatedly. Prepared cell lysates were evaluated by SDS-PAGE under reducing conditions and WB using in-house-generated mouse MAb 6C2 against His-tag at 3 μg/mL concentration or CMY-34 specific MAbs at 2 μg/mL concentration.

## Sequencing of MAb variable regions

Determination of MAb variable heavy chain (VH) and light chain (VL) sequences was performed as described previously (*Stravinskiene et al., 2020*). Briefly, total RNA was extracted from hybridoma cells using GeneJET RNA Purification Kit (K0732; Thermo Scientific, Waltham, MA, USA), following the manufacturer's protocol. Reverse transcription was carried out using the extracted RNA as a template and the RevertAid First Strand cDNA Synthesis Kit (K1621; Thermo Scientific, Waltham, MA, USA), according to the manufacturer's recommendations. The variable regions were amplified by PCR using the Phusion Flash High-Fidelity PCR Master Mix (F548S; Thermo Scientific, Waltham, MA, USA) and previously described primers (Table 3) (*Pleckaityte et al., 2011*). Amplified DNA fragments were cloned using CloneJET PCR Cloning Kit (K1232; Thermo Scientific, Waltham, MA, USA) and verified by Sanger sequencing (GENEWIZ). The results were analyzed with the IgBlast tool of the National Center for Biotechnology Information (NCBI, https://www.ncbi.nlm.nih.gov/igblast/) (*Ye et al., 2013*), and VH and VL coding nucleotide sequences were identified. The sequences of complementarity-determining regions 1–3 (CDR1–3) were determined using a tool for

standardized analysis of the antibody sequences IMGT/V-QUEST version 3.6.1 (https://www.imgt.org/IMGT_vquest/input) (*Brochet, Lefranc & Giudicelli, 2008*; *Giudicelli, Brochet & Lefranc, 2011*).

### Lateral flow immunoassay (LFIA)

Lateral flow immunoassay (LFIA) test strips were assembled of a nitrocellulose (NT) membrane Vivid 120 (VIV120SAMP; Cytiva, Marlborough, MA, USA), a 6,613 polyester fiber conjugate pad (Ahlstrom-Munksjo, Espoo, Finland), a 1,281 cotton sample pad (Ahlstrom-Munksjo, Espoo, Finland), a grade 243 wick pad (Ahlstrom-Munksjo, Espoo, Finland) and a self-adhesive base (nanoComposix, San Diego, CA, USA). The colloidal gold-conjugated MAb was prepared by covalent conjugation with BioReady$^{TM}$ 40 nm Carboxyl Gold spheres (AUXR40-5M; nanoComposix, San Diego, CA, USA), according to the manufacturer's recommendations. The conjugate pad was prepared by its immersion in the treatment buffer (10% sucrose (w/v), 10 mM potassium phosphate, pH 7.4) for 10 min, followed by incubation with the gold-MAb conjugate (25 $\mu L/cm^2$) for 1 h at RT, then drying for 5 h at 37 °C, followed by drying with desiccator overnight at RT. The test zone was prepared by immobilizing MAbs at 1 mg/mL concentration that were mixed with green dye: ProClin$^{TM}$ 300 (48912-U; Sigma-Aldrich, Burlington, MA, USA), dilution factor 1:40 with PBS, 3.3 mM Orange G (O7252; Sigma-Aldrich, Burlington, MA, USA), 0.56 mM xylene cyanole (B3267; Sigma-Aldrich, Burlington, MA, USA), dilution factor 1:10 with PBS. The control zone was formed by 0.2 mg/mL goat anti-mouse IgG Fc antibodies (103301; SouthernBiotech, Birmingham, AL, USA) mixed with blue dye (3.7 mM xylene cyanole, dilution factor 1:20 with PBS). The contour zone consisted of BSA conjugated with Remazol Brilliant Blue R dye (R8001; Sigma-Aldrich, Burlington, MA, USA). The antibodies were spotted onto the membrane (drop volume 21.6 nL) by the program-controlled dispensing system sciFLEXARRAYER S3 (Scienion, Volmerstraße, Germany), and then dried for 1 h at 37 °C. The sample pad was treated with blocking solution (2% BSA (w/v) in PBS) for 30 min at RT, rinsed two times with distilled water, incubated with treatment buffer for 30 min at RT, and dried for 1 h at 37 °C.

Before testing, bacterial isolates were grown on LB agar supplemented with 16 µg/mL ceftazidime for 20–24 h at 30 °C. For testing, a single colony was resuspended in 150 µL of extraction buffer (1% (w/v) 3-((3-cholamidopropyl)-dimethylammonium)-1-propanesulfonate (CHAPS, Carl Roth, Karlsruhe, Germany, 1479.1), 0.5% (w/v) BSA, 0.5% (v/v) Tween-20 in 1:1 diluted PBS) and vortexed. A total of 100 µL of sample was dispensed onto the sample pad, and the results were evaluated visually after 20 min.

### Two-photon excitation (TPX) assay

Performance of MAbs against CMY-34 was evaluated in separation-free double sandwich immunoassay utilizing two-photon excitation fluorescence detection (*Hänninen et al., 2000*; *Koskinen et al., 2021*), as applied in commercial mariPOC® test (ArcDia International Ltd., Turku, Finland). MAbs 9D2 and 2E11 were coated on microparticles, labelled with fluorescent molecules, and mixed together to obtain assay reagent cocktail as described previously (*Koskinen et al., 2007*).

For determination of analytical sensitivity of the assay, rCMY-34 was serially diluted with assay buffer (R-B01; ArcDia International Ltd, Turku, Finland) in the range of 0.5–1,500 ng/mL. For testing, bacterial isolates were grown on blood agar for 24 h at 30 °C. A single colony was suspended in 100 µL of extraction buffer (1% CHAPS, 0.5% (w/v) BSA, 0.5% (v/v) Tween-20 in 1:1 diluted PBS) and diluted with assay buffer. Diluted rCMY-34 samples and bacterial suspensions were mixed with the assay reagent cocktail and analyzed.

## Testing of bacterial isolates by PCR

Bacterial isolates used for MAb characterization were additionally tested by PCR using previously described primers specific to $bla_{CMY}$ (CMY-F1 5′-MTGGGGKAAAGCCG ATATC-3′ and CMY-R1 5′-AGTTCAGCATCTCCCANCC-3′) (*Mlynarcik et al., 2021*) and $bla_{NDM}$ (NDM-F 5′-GGGGATTGCGACTTATGC-3′ and NDM-R 5′-AGATTGCCG AGCGACTTG-3′) (*Mlynarcik et al., 2019*). During the PCR, a 701 bp fragment of $bla_{CMY}$ and 258 bp fragment of $bla_{NDM}$ were amplified. Obtained PCR products of isolates with previously unidentified β-lactamases were sequenced and analyzed by the Beta-Lactamase DataBase (BLDB) BLAST tool (http://www.bldb.eu:4567/) (*Naas et al., 2017*).

## Bacterial isolates

In total, nine bacterial isolates with known β-lactamase profiles and two β-lactamase-negative isolates were tested. The isolates used for MAb characterization and LFIA testing were: CMY-34-positive *Citrobacter portucalensis* (3826Z08), which was a gift from Anette M. Hammerum and colleagues (*Hammerum et al., 2011*), NDM-1-positive *Klebsiella pneumoniae* (CCUG60138 and CCUG68728), *Acinetobacter chinensis* (CCUG74036T and CCUG74037), which were obtained from the Culture Collection University of Gothenburg (Sweeden). The isolates tested by LFIA and TPX assays were: CMY-6, NDM-1, OXA-1, CTX-M-15, SHV-11-positive *K. pneumoniae* (50627996), CMY-4, OXA-1, VIM-29, CTX-M-15-positive *E. coli* (50639799), CMY-16, NDM-1, OXA-10-positive *Proteus mirabilis* (50664164), CMY-2, OXA-1, OXA-181, CTX-M-15, TEM-1-positive *E. coli* (50816743), β-lactamase-negative *Klebsiella aerogenes* (50796520) and *P. mirabilis* (50793946), which were provided by Clinical microbiology department of Turku University Hospital (Finland).

## Preparation of bacterial lysates

For preparation of lysates, *K. pneumoniae* (CCUG60138, CCUG68728) and *A. chinensis* (CCUG74036T, CCUG74037) isolates were grown in LB media supplemented with 16 µg/mL meropenem and *C. portucalensis* (3826Z08) isolate was grown in LB supplemented with 16 µg/mL ceftazidime for 16–20 h at 30 °C. The cells were disrupted by sonication, and the suspension was cleared by centrifugation at 15,000× *g* for 20 min at 4 °C. Total protein concentration was determined using Pierce™ Bradford Protein Assay Kit (23200; Thermo Scientific, Waltham, MA, USA), according to the manufacturer's protocol. The supernatants were aliquoted and stored at −70 °C.

## Cefinase test

The enzymatic activity of β-lactamases was tested with chromogenic cephalosporin (nitrocefin) impregnated BBL™ Cefinase™ paper discs (BD Biosciences, Franklin Lakes, NJ, USA). The discs were dispensed into sterile petri dishes. Then, 2.5 μg of rCMY-34 or 5 μg of bacterial lysate with known total protein concentration was diluted with PBS to the total volume of 20 μL. The samples were dispensed onto the discs, incubated for 1 h at RT, and evaluated visually.

## Structure prediction of MAb-antigen complex

Sequences of MAb Fab fragment were constructed by concatenating determined MAb 2E11 and 9D2 VH and VL sequences with the corresponding UniProtKB (*Bateman et al., 2023*) sequences of mouse IgG2a Ig-like domain 1 for the 2E11 heavy chain (UniProt accession no. P01863), mouse IgG1 sequences for 9D2 heavy chain (UniProt accession no. P01868) and mouse immunoglobulin kappa for the light chain (UniProt accession no. P01837).

Structure modeling of the interaction between CMY-34 β-lactamase (GenBank accession no. ABN51006.1) and Fab-variable fragments of the MAbs was performed using AlphaFold2-Multimer (*Jumper et al., 2021*; *Evans et al., 2022*). In the first step, models were generated using standard AlphaFold and ColabFold pipelines as described previously (*Olechnovič et al., 2023*). No reliable models were produced, therefore the AFsample protocol was used to generate 6,000 models (*Wallner, 2023*). AFsample models having self-assessment score (ranking_confidence) higher than 0.8 were relaxed using OpenMM (*Eastman et al., 2017*). In addition to the AlphaFold quality self-estimation, the antibody-antigen interfaces in the predicted structures were also evaluated by the VoroIF-jury procedure using the FTDMP software (*Olechnovič et al., 2023*, *2025*), as well as VoroMQA (*Olechnovič & Venclovas, 2017*) and VoroIF-GNN (*Olechnovič & Venclovas, 2023*) scores. For further analysis of MAb 2E11 epitope, the model having minimum predicted-aligned error (PAE) value for the antibody-antigen interface was selected as the most reliable model.

Additionally, we decided to use AlphaFold3 (*Abramson et al., 2024*) for the antibody-antigen interaction modeling. The models were generated using the default parameters of AlphaFold Server with or without the templates, and the most reliable model was selected according to the reported ranking score values.

The protein interaction interfaces were compared using contact area difference (CAD)-score (*Olechnovič & Venclovas, 2020*). The contacts between antibody and antigen molecules were analyzed using the VoroContacts server (*Olechnovič & Venclovas, 2021*).

## Statistical analysis and visualization of the data

The data was analyzed, and the graphs were generated using GraphPad Prism (Dotmatics, Boston, MA, USA) and OriginPro 8 (OriginLab) softwares. The ELISA results of MAb reactivity testing with β-lactamases and CMY-34 fragments are presented as mean of OD values ± standard deviation (SD) of three replicates ($n = 3$). The cut-off values were calculated as sum of three SD (3SD) of negative control. The ELISA results of MAb testing

with synthetic peptides are demonstrated as mean ± SD, $n = 3$. The indirect ELISA results of MAb cross-reactivity testing are presented as mean ± SD, $n = 3$, cut-off value of negative control. Results of developed sandwich ELISA are presented as mean ± SD, $n = 3$, cut-off value of negative control (*E. coli* BL21 lysate). The results of dose-response curve and testing of bacterial isolates by TPX assay are shown as mean, $n = 3$, cut-off value of negative control. $K_d$ values were calculated with OriginPro 8 (OriginLab) software, and the results are demonstrated as mean ± standard error of mean (SEM), $n = 3$. The limit of detection of the assays was defined as 3SD from intercept as described previously (*Armbruster & Pry, 2008*).

## RESULTS

### Generation of MAbs against CMY-34 and determination of their specificity and affinity

For generation of MAbs, recombinant His-tagged β-lactamase CMY-34 (rCMY-34) was expressed in *E. coli* and purified by nickel-affinity chromatography (Fig. S1). The enzymatic activity of the purified protein was confirmed by cefinase test (Fig. S2). For this purpose, rCMY-34 was applied on the paper disc that is impregnated with chromogenic cephalosporin—nitrocefin. During the testing, nitrocefin turned from yellow to red, indicating the hydrolysis of this β-lactam and the proper folding of the recombinant enzyme.

In the next step, rCMY-34 was used for immunization of mice and generation of CMY-specific MAbs. Following hybridization, more than 130 hybridoma clones were selected in total for further testing of MAb reactivity with the antigen, of which fewer than a half were cloned and had their isotype determined. Following the screening, the collection of 14 IgG class MAbs was created (Table S1). To evaluate the binding affinity of generated MAbs with rCMY-34, their apparent dissociation constants ($K_d$) were determined by indirect ELISA (Table S1). Experimentally obtained $K_d$ values of the MAbs were in the range of 0.05–2.5 nM, indicating high-affinity interaction with rCMY-34 (Table S1). In the following step, noncompeting MAb pairs suitable for sandwich-type assays were selected. For this purpose, six high affinity MAbs were conjugated with HRP and tested by competitive ELISA (Fig. S3). Selected candidate pairs of MAbs were additionally tested by sandwich ELISA to evaluate their ability to detect rCMY-34 with the highest sensitivity (Fig. S4). Based on these results, two non-competing, high-affinity MAbs produced by hybridoma clones 9D2 (IgG1, $K_d$ 0.24 nM) and 2E11 (IgG2a, $K_d$ 0.10 nM) were selected for further investigation by a wide range of immunoassays (Table S1, Fig. S5). Their reactivity with rCMY-34 was additionally confirmed by indirect ELISA, western blot (WB) and immunoprecipitation (IP) methods (Fig. 1, Table 4).

To further characterize the reactivity of selected MAbs and their ability to recognize β-lactamases, several antibiotic-resistant β-lactamase producing bacterial isolates were analyzed. In the first step, a previously described (*Hammerum et al., 2011*) *C. portucalensis* isolate (3826Z08) producing CMY-34 β-lactamase was tested. The presence of CMY-34

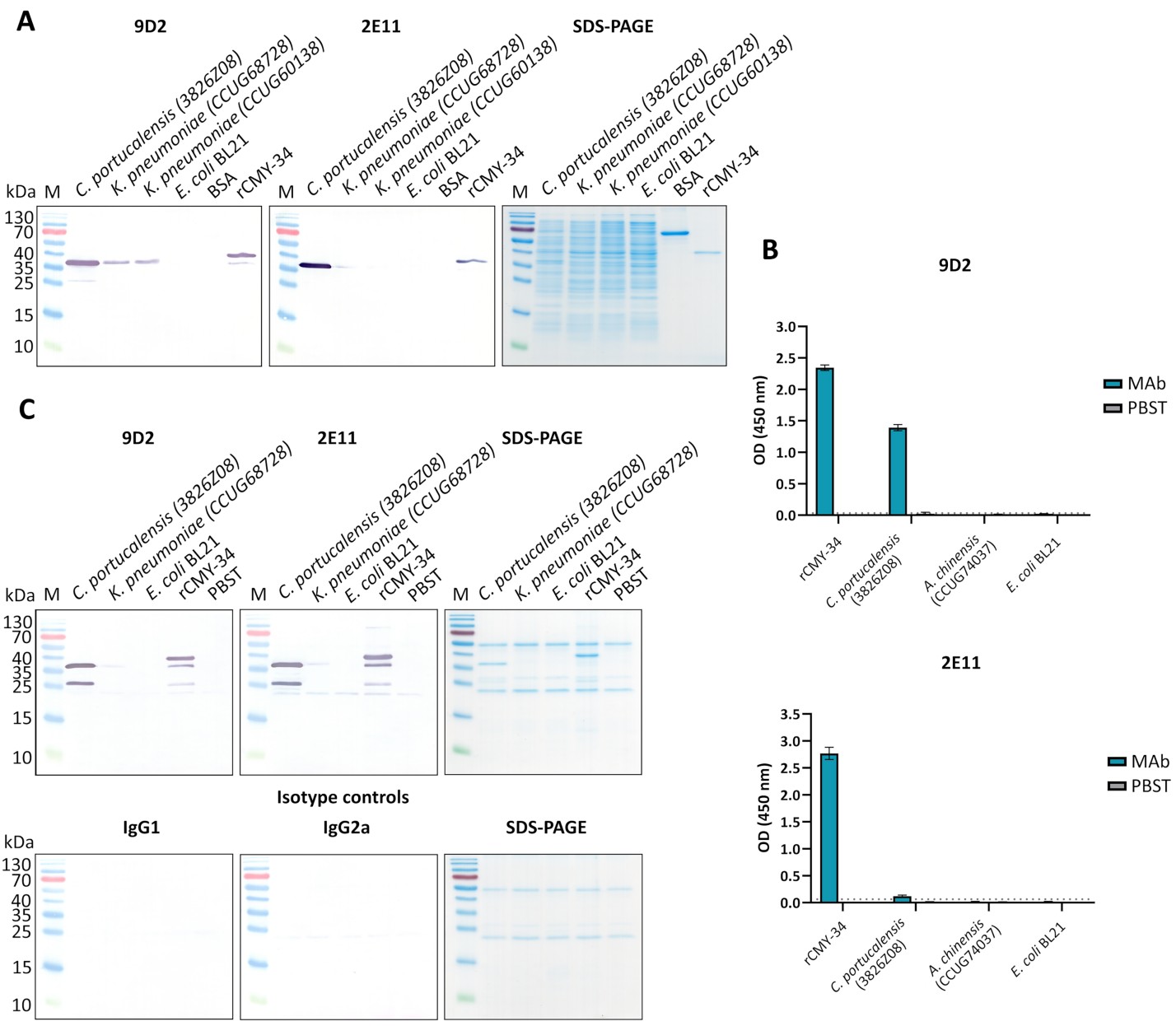

**Figure 1 The reactivity of MAbs 9D2 and 2E11 against CMY-34 with β-lactamases tested by different immunoassays.** The lysates of CMY-34 producing *C. portucalensis* and CMY-4 producing *K. pneumoniae* were used as a source of CMY β-lactamases. Lysates of *E. coli* BL21 and NDM-1 producing *A. chinensis* were tested as negative control. (A) MAb reactivity with β-lactamases tested by WB. BSA, bovine serum albumin. (B) MAb reactivity with β-lactamases tested by indirect ELISA ($n = 3$, mean ± SD, dashed lines represent the cut-off values 0.035 and 0.064 for MAbs 9D2 and 2E11, respectively). (C) WB analysis of immunoprecipitated β-lactamases. *E. coli* BL21 lysate and protein dilution buffer (PBST) were tested as negative controls. In-house-produced irrelevant IgG1 MAb 18E4 against house dust mite allergen Der p 23 and IgG2a MAb 20G11 against SARS-CoV-2 spike protein were analyzed as isotype controls. HRP-conjugated MAb 9D2 was used as detection antibody. SDS-PAGE results of immunoprecipitated proteins with MAbs 9D2 and isotype control 18E4 are demonstrated. M—PageRuler Prestained Protein Ladder (Thermo Scientific, 26616).

coding gene was additionally confirmed by PCR, with a set of previously described primers (*Mlynarcik et al., 2021*) targeting a 701 bp region of *bla*$_{CMY}$. Moreover, the PCR analysis of two NDM-1 β-lactamase producing *K. pneumoniae* isolates (CCUG60138 and

**Table 4 Summarized results of MAbs 9D2 and 2E11 raised against rCMY-34 characterization.**

| MAb clone | Assay | MAb reactivity and cross-reactivity with β-lactamases | | | |
|---|---|---|---|---|---|
| | | rCMY-34 | CMY-34 | CMY-4 | rPDC-195[b] |
| 9D2 | ELISA | + | + | + | + |
| | WB[a] | + | + | + | + |
| | IP[a] | + | + | + | Not tested |
| 2E11 | ELISA | + | + | + | – |
| | WB | + | + | + | – |
| | IP | + | + | + | Not tested |

Notes:
[a] Visible band was defined as positive result (+), the absence of visible band was considered as negative result (–) in WB.
[b] Recombinant PDC-195 β-lactamase.

CCUG68728) with $bla_{CMY}$ and $bla_{NDM}$ specific primers (*Mlynarcik et al., 2019, 2021*) revealed that these isolates also encode a CMY β-lactamase (Fig. S6). The sequencing of amplified PCR products confirmed that both *K. pneumoniae* isolates contained $bla_{CMY-4}$. Therefore, these isolates were used as CMY-4 producers for further characterization of MAbs and the development of CMY-specific immunoassays. As a negative control for these assays, *E. coli* BL21 strain and two NDM-1 producing *A. chinensis* isolates (CCUG74036T and CCUG74037) were utilized.

The ability of MAbs to recognize CMY β-lactamases was tested using lysates of above described CMY-34, CMY-4 and NDM-1 producing bacterial isolates with known total protein concentration. After preparation of lysates, the retained enzymatic activity of β-lactamases was confirmed by cefinase test (Fig. S2). The analysis revealed nitrocefin hydrolysis, indicating that the lysate preparation conditions did not disrupt structure of β-lactamases, which is important for their catalytic activity. The testing of *C. portucalensis* (3826Z08) lysate by WB, IP and indirect ELISA confirmed the reactivity of MAbs 9D2 and 2E11 with CMY-34 (Fig. 1, Table 4). Moreover, cross-reactivity of MAbs 9D2 and 2E11 with CMY-4 was revealed by WB and IP when lysates of $bla_{CMY-4}$ harboring *K. pneumoniae* (CCUG60138 and CCUG68728) were analyzed (Fig. 1, Table 4). Protein bands of approximately 38–39 kDa corresponding to CMY-34 and CMY-4 in the respective bacterial lysates were identified by WB, along with a 40 kDa band of rCMY-34, which was used as a positive control (Figs. 1A, 1B). Furthermore, both MAbs immunoprecipitated CMY-34 and CMY-4 from *C. portucalensis* (3826Z08) and *K. pneumoniae* (CCUG60138 and CCUG68728) lysates, respectively (Fig. 1C, Table 4). The distinct protein bands visible in WB below the full-length natural CMY-34 in the *C. portucalensis* lysate and recombinant CMY-34 may correspond to partially degraded forms of CMY-34 that still retain MAb recognition site (Fig. 1C). Thus, due to their high affinity, MAbs 9D2 and 2E11 were able to capture and immunoprecipitate all forms of these β-lactamases—both full-length and partially degraded. Protein band observed in WB slightly below the 25 kDa corresponds to the light chain of the MAb used for IP (Fig. 1C). Considering this, a control IP reaction with only IP MAb (no cell lysate or target proteins) was analyzed to identify bands derived solely from the antibody (Fig. 1C, lane PBST).

This control demonstrates that a protein band corresponding to the light chain was derived from the antibody used for IP.

## Determination of MAb variable sequences

Variable regions of heavy (VH) and light (VL) chains of the MAbs 9D2 and 2E11 were determined. Total RNA isolated from the hybridoma cells (Fig. S7A) was used to synthesize single-stranded cDNA. The VH and VL coding sequences were amplified by PCR using primers specific to mouse immunoglobulin gene heavy and light chains localized in the framework region 1 (*Orlandi et al., 1989*; *Wang et al., 2000*; *Pleckaityte et al., 2011*) (Figs. S7B, S7C). The amplified PCR products were cloned into the cloning vector, sequenced and analyzed with the IgBlast (https://www.ncbi.nlm.nih.gov/igblast/) (*Ye et al., 2013*) and IMGT/V-QUEST (https://www.imgt.org/IMGT_vquest/input) (*Brochet, Lefranc & Giudicelli, 2008*; *Giudicelli, Brochet & Lefranc, 2011*) tools. After analysis, the VH and VL coding sequences and the sites of complementarity-determining regions 1–3 (CDR1–3) were determined. The obtained VH and VL sequences confirmed the presence of clonal differences between MAbs 9D2 and 2E11. The sequences are doposited in the ABCD (AntiBodies Chemically Defined) Database (*Lima et al., 2020*) with assigned accession numbers ABCD_BD767 and ABCD_BD768 for MAb 9D2 and 2E11, respectively.

## Mapping of MAb epitopes

It was demonstrated, that selected pair of MAbs 9D2 and 2E11 do not compete for rCMY-34 binding (Fig. S3), which is indicative of their distinct epitopes. In the next step, analysis of MAb recognition sites within the CMY-34 β-lactamase was performed. For this purpose, six overlapping His-tagged CMY-34 fragments (CMY-34_1–CMY-34_6) were expressed in *E. coli* (Fig. 2A). The efficiency of protein expression was confirmed by SDS-PAGE and WB when *E. coli* lysates after protein synthesis induction were analyzed (Fig. S8). For epitope mapping, the same *E. coli* lysates were tested with MAbs 9D2 and 2E11 by WB and sandwich ELISA on microtiter plates coated with in house-generated His-tag-specific MAb 6C2 (Figs. 2B, 2C). Both ELISA and WB revealed the reactivity of MAb 9D2 with the overlapping fragments CMY-34_1 and CMY-34_2, indicating MAb 9D2 epitope localization at 22–49 amino acids (aa) of CMY-34 (according to CMY-34 sequence: GenBank accession no. ABN51006.1), as this region is present in both CMY-34_1 and CMY-34_2 fragments but absent in the non-reactive CMY-34_3 fragment (Table 2). However, it was determined that MAb 2E11 did not react with any of CMY-34 fragments, according to ELISA and WB results (Figs. 2B, 2C), suggesting that epitope is not strictly linear and is disrupted by dividing the CMY-34 protein into the fragments.

For fine epitope mapping of MAb 9D2, the overlapping synthetic peptides (P0–P16) spanning a determined MAb 9D2 epitope (22–49 aa region of CMY-34) were analyzed (Fig. 3A). MAb 9D2 reactivity with synthetic peptides was evaluated by indirect ELISA, which revealed the reactivity of the MAb with P0 (22–49 aa region of CMY-34 coresponding peptide) and P9–P13 peptides (Fig. 3B). Based on these results, MAb 9D2

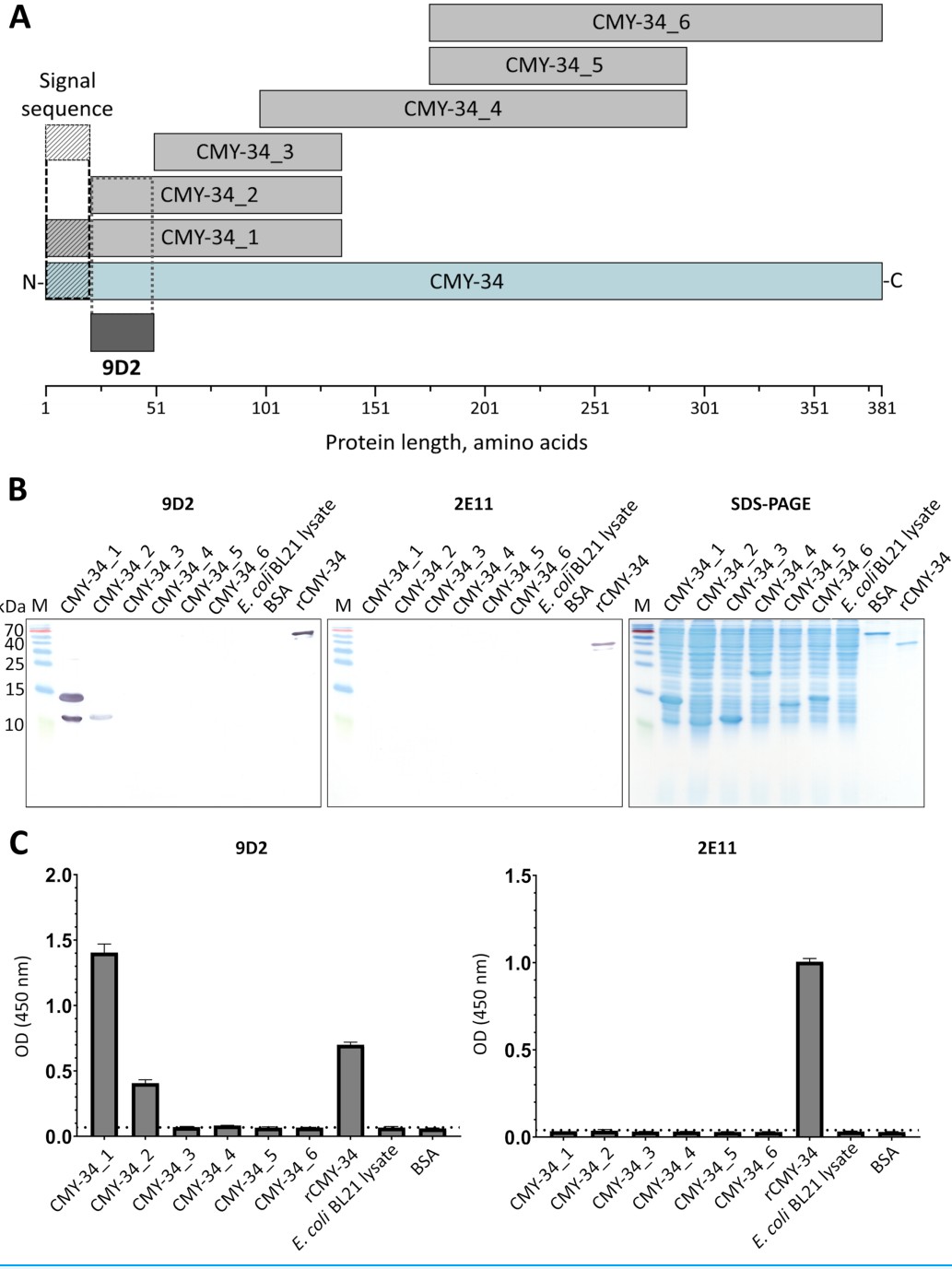

**Figure 2 MAb testing with CMY-34 fragments (CMY-34_1–CMY-34_6).** *E. coli* lysates after synthesis induction of CMY-34 fragments were analyzed. *E. coli* BL21 lysate and bovine serum albumin (BSA) were tested as negative controls. (A) Schematic representation of the overlapping CMY-34 fragments used for epitope mapping. The fragments are colored light gray and the full-length CMY-34 protein is colored cyan. Determined recognition site of MAb 9D2 is underlined and colored dark gray. The signal sequence of CMY-34 protein is underlined separately. (B) MAb reactivity with CMY-34 fragments tested by WB. M—PageRuler Prestained Protein Ladder (Thermo Scientific, 26616). (C) MAb reactivity with CMY-34 fragments tested by sandwich ELISA ($n = 3$, mean ± SD, dashed lines represent the cut-off values 0.065 and 0.036 for MAbs 9D2 and 2E11, respectively). The microtiter plates were coated with in house-generated His-tag-specific MAb 6C2, which was used as capture antibody. *E. coli* BL21 lysate and BSA were tested as negative controls.

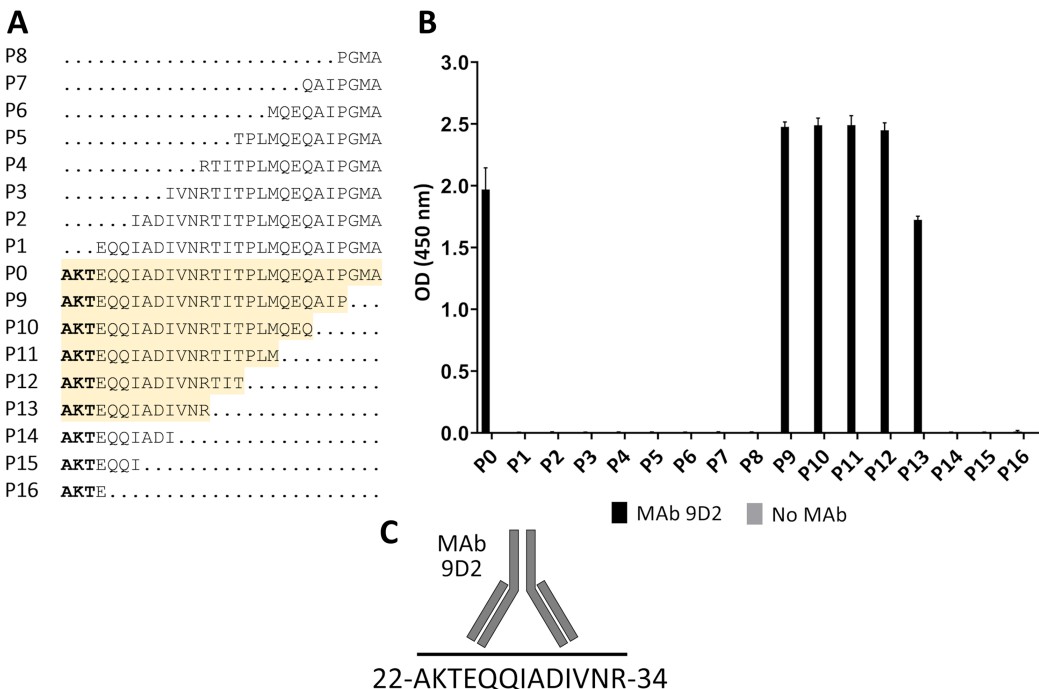

**Figure 3 Mapping of MAb 9D2 epitope with synthetic peptides (P0–P16) spanning 22–49 aa region of CMY-34.** (A) The aa sequences of overlapping synthetic peptides used for epitope mapping. P0 peptide corresponds to previously determined MAb 9D2 recognition site at 22–49 aa of CMY-34. The peptides that showed MAb reactivity are highlighted in yellow. (B) The indirect ELISA results of MAb reactivity testing with synthetic peptides ($n = 3$, mean ± SD). As a negative control, the wells without the MAb were tested. (C) Graphic visualization of determined MAb 9D2 epitope localized at 22–34 aa of CMY-34.

epitope consists of 13 aa-long segment, which is localized at 22–34 aa region of CMY-34 (according to CMY-34 sequence: GenBank accession no. ABN51006.1) (Fig. 3C).

## Computational prediction of MAb 2E11 epitope

Since the localization of MAb 2E11 epitope within the CMY-34 protein has not been experimentally determined using truncated CMY-34 fragments, computational modeling of CMY-34 and MAb 2E11 interaction was performed to determine the putative epitope. The modeling was performed using the sequences of CMY-34 and MAb 2E11 Fab fragment. In total, 6000 AlphaFold2 (*Jumper et al., 2021*; *Evans et al., 2022*; *Wallner, 2023*) models were generated. Only 20 models had high self-assessment score (ranking_confidence >0.8, Fig. S9), and 19 of them were highly similar to each other, having an average antibody-antigen interface CAD-score value of 0.82 (Fig. S10A). Interestingly, this binding mode was also observed as a top ranked AlphaFold model when only the VH and VL fragments of the MAb were used for modeling of interaction (Fig. S10B) and in the top ranked AlphaFold3 (*Abramson et al., 2024*) model. These 19 models were also the most reliable models when evaluating them with VoroIF-jury (*Olechnovič et al., 2023*, *2025*), which is a protocol independent of AlphaFold quality self-estimation. The evaluation suggested that the prediction was accurate enough for epitope

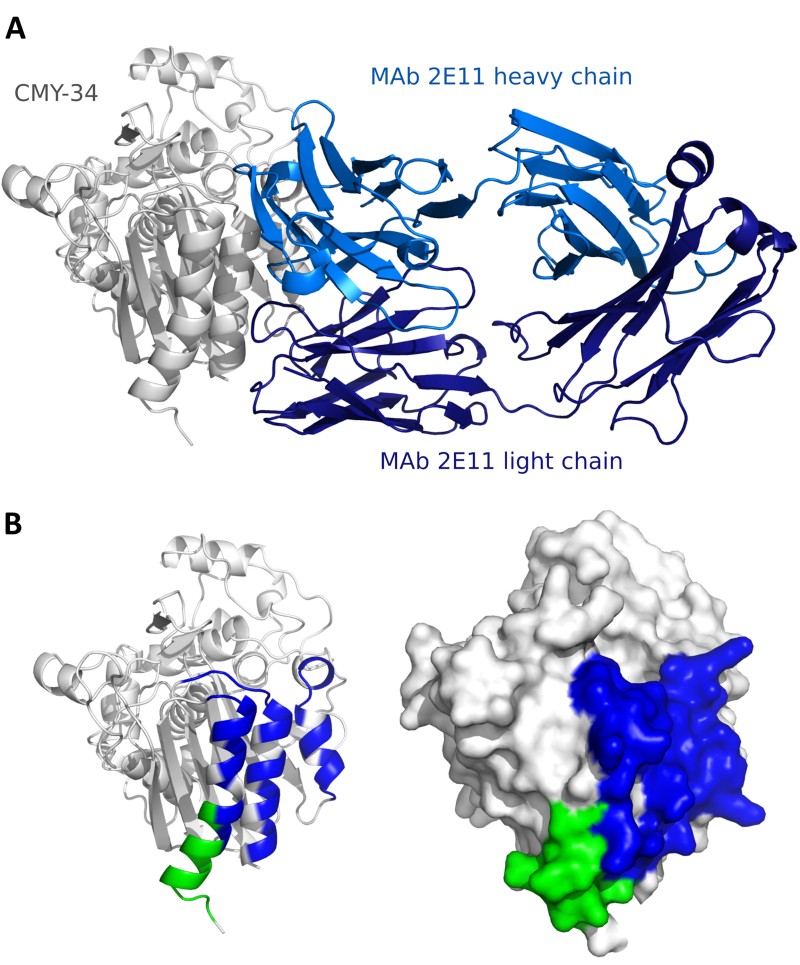

**Figure 4 Predicted models of interaction between CMY-34 and MAb 2E11.** (A) Structure model of CMY-34 (colored gray, signal sequence residues (1–20 aa) are not demonstrated for clarity) and MAb 2E11 (colored blue) interaction. (B) Localization of MAb 9D2 (determined experimentally, colored green) and MAb 2E11 (predicted, colored blue) epitopes visualized on CMY-34 structure model (signal sequence residues (1–20 aa) are not demonstrated).

estimation, therefore, a model having minimal average PAE value at the MAb-CMY-34 interface was selected for further analysis (Fig. 4A). According to the most reliable AlphaFold model, the putative MAb 2E11 epitope is conformational and consists of three short segments located at 38–44, 299–311 and 362–379 aa of CMY-34 (according to CMY-34 sequence: GenBank accession no. ABN51006.1) (Fig. S11). Numerous hydrogen bonds are present between polar residues of antibody and antigen in every fragment of the interaction interface, and there are several possible salt bridges identified (Fig. S10C).

The spatial arrangement of the putative epitope of MAb 2E11 is in close proximity to the MAb 9D2 recognition site (Fig. 4B). AlphaFold2-based AFsample protocol did not generate any reliable models of the MAb 9D2 and CMY-34 interaction (Fig. S9), but in the AlphaFold3 model of CMY-34 and MAb 9D2 complex, the antibody is bound to the antigen in different possition than MAb 2E11 (Fig. S12). The overlap between the epitopes

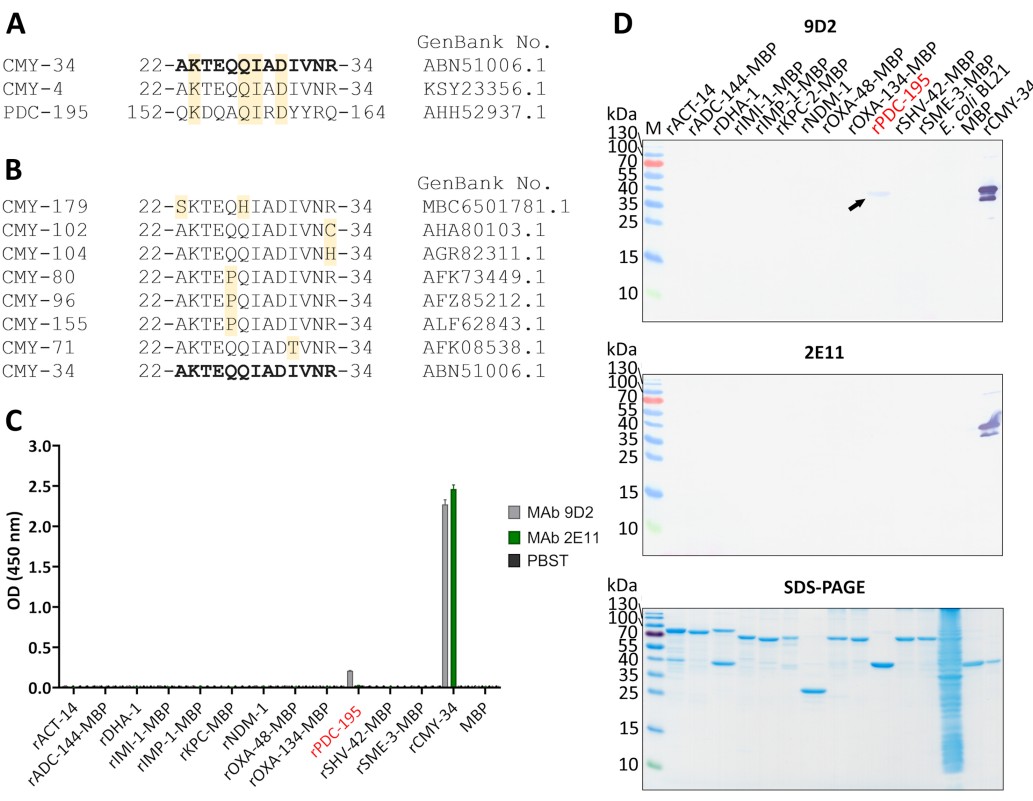

**Figure 5 Cross-reactivity analysis of MAbs 9D2 abd 2E11.** (A) Alignment of MAb 9D2 recognition site with the respective sequences of cross-reactive β-lactamases CMY-4 (22–34 aa region) and PDC-195 (152–164 aa region). The positions of matched aa are colored yellow. CMY-34 corresponding sequence is bolded. Sequences were aligned with Clustal Omega (*McWilliam et al., 2013*). (B) Alignment of CMY β-lactamases with identified aa differences within the MAb 9D2 epitope (22–34 aa region of CMY-34). The positions of mismatched aa are marked in yellow. CMY-34 coresponding sequence is bolded. (C) The results of MAbs 9D2 and 2E11 cross-reactivity with recombinant β-lactamases tested by indirect ELISA (*n* = 3, mean ± SD, dashed line represents a cut-off value 0.013 for MAbs 9D2 and 2E11). Cross-reactive rPDC-195 is colored red. Maltose binding protein (MBP) was tested as negative control. (D) MAb cross-reactivity testing by WB. The arrow indicates the position of positive band in WB when MAb 9D2 was tested with rPDC-195 (indicated in red). *E. coli* BL21 lysate and MBP were used as negative controls. M—PageRuler Prestained Protein Ladder (Thermo Scientific, 26616).

is low, thus the results of structure modeling are in line with the experimentally observed independence of the epitopes.

## Analysis of MAb cross-reactivity

The identified binding site of MAb 9D2 (22–34 aa region of CMY-34) was further analyzed by Beta-Lactamase DataBase (BLDB) BLAST tool (http://www.bldb.eu:4567/) (*Naas et al., 2017*) to determine its similarity across the members of CMY family. The analysis revealed 100% sequence identity among 173 of 181 currently identified enzymes of the CMY family, including in this study analyzed CMY-4 variant (Fig. 5A). The remaining seven members of CMY family (CMY-71, CMY-80, CMY-96, CMY-102, CMY-104, CMY-155, CMY-179)

had only one or two aa differences within this region (Fig. 5B) (according to BLDB, last update of the database: January 22, 2025).

The sequence analysis of putative MAb 2E11 recognition site was subsequently performed. The alignment of MAb 2E11 binding regions (38–44, 299–311 and 362–379 aa of CMY-34) with CMY β-lactamase family, revealed high sequence homology to most of the CMY enzymes. It was determined that 38–44 aa region of CMY-34 was identical to all identified members of CMY family. Moreover, there were only one or two aa mismatches in 19 CMY variants observed, when 299–311 aa region of CMY-34 was aligned (Table S2). Analysis of 362–379 aa region revealed that 48 CMY variants had one aa and five CMY variants had two aa differences in this segment (Table S3).

For further analysis of MAb cross-reactivity, MAbs 9D2 and 2E11 were tested with 12 recombinant β-lactamases by indirect ELISA and WB (Figs. 5C, 5D). Selected recombinant enzymes represent different classes of β-lactamases. According to Ambler's classification (*Ambler, 1980*), for cross-reactivity testing used recombinant IMI-1, KPC-2, SHV-42 and SME-3 β-lactamases correspond to class A, recombinant IMP-1 and NDM-1—class B, recombinant ACT-14, ADC-144, DHA-1, PDC-195—class C, recombinant OXA-48 and OXA-134 variants—class D β-lactamases. After analysis, weak cross-reactivity of MAb 9D2 with recombinant PDC-195 (rPDC-195) β-lactamase was determined by both WB and ELISA (Figs. 5C, 5D). The alignment of MAb 9D2 epitope (22–34 aa region of CMY-34) with PDC-195 aa sequence revealed a 152–164 aa region of PDC-195 containing one or two aa segments identical to those of CMY-34 and CMY-4 (Fig. 5A). In contrast to MAb 9D2, there was no cross-reactivity of MAb 2E11 with rPDC-195 and the remaining recombinant β-lactamases observed (Figs. 5C, 5D).

## Application of MAbs for development of immunoassays

Characterized MAbs 9D2 and 2E11 were applied in sandwich ELISA and rapid analytical tests, such as lateral flow immunoassay (LFIA) and two-photon excitation (TPX) assay (*Koskinen et al., 2007*), for immunodetection of CMY β-lactamases in bacterial isolates.

In the first step, the ability of MAbs to capture and detect CMY enzymes in bacterial lysates was evaluated by sandwich ELISA. For testing and optimization of the assay, the microtiter plates were coated with MAb 2E11 for antigen capture, while HRP-conjugated MAb 9D2 was utilized as detection antibody, and rCMY-34 was tested as target antigen. The optimal plate type, MAb concentration, immobilization and blocking conditions showing the highest sensitivity of the assay were selected. A standard calibration curve with rCMY-34 was generated (Fig. 6A), and observed limit of detection for rCMY-34 was 43 pg/mL. Moreover, the developed sandwich ELISA was tested for detection and quantification of CMY β-lactamases in *C. portucalensis* (3826Z08, CMY-34 producer) and *K. pneumoniae* (CCUG60138 and CCUG68728, CMY-4 and NDM-1 producers) lysates with known total protein concentration of 555 μg/mL. Lysates of *E. coli* BL21 strain and *A. chinensis* isolate (CCUG74036T, NDM-1 producer) were analyzed as negative controls (Fig. 6B). The determined CMY-34 concentration in the *C. portucalensis* (3826Z08) lysate

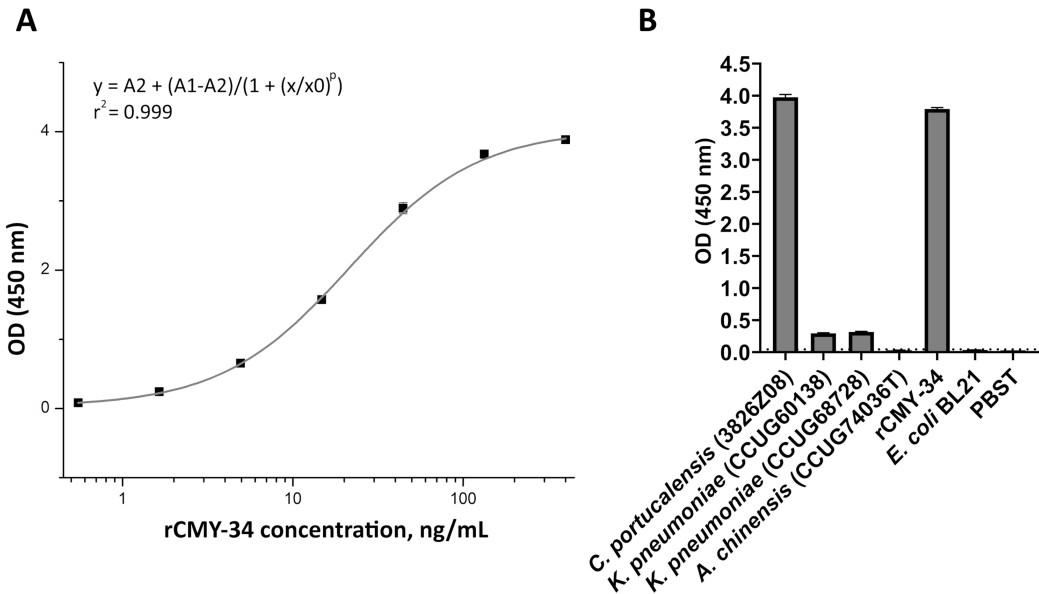

**Figure 6** **Application of MAbs 9D2 and 2E11 in sandwich ELISA for detection of CMY β-lactamases.** (A) The calibration curve generated with the capture MAb 2E11, HRP-conjugated detection MAb 9D2 and rCMY-34 as a standard. (B) Application of sandwich ELISA for detection of CMY β-lactamases in bacterial lysates. The lysates of CMY-34 producing *C. portucalensis* and CMY-4 producing *K. pneumoniae* were tested. The lysates of *E. coli* BL21, NDM-1 producing *A. chinensis*, and protein dilution buffer (PBST) were tested as negative controls. The bars represent OD values (*n* = 3, mean ± SD) when bacterial lysates with total protein concentration of 555 μg/mL were analyzed. The dashed line indicates a cut-off value of 0.043.

was 34.5 ng/mL. Meanwhile, CMY-4 concentration in two analyzed *K. pneumoniae* (CCUG60138 and CCUG68728) lysates were 9.3 and 10.6 ng/mL, respectively.

In the next step, MAbs 9D2 and 2E11 were applied in rapid analytical test, such as LFIA. For the assay, MAb 2E11 was immobilized on the nitrocellulose membrane and used as a capture antibody. The MAb 9D2 was labeled with 40 nm gold nanoparticles and used for visual detection. The combination of materials used for assembly of the test strips and concentration of MAbs showing the highest sensitivity were selected. Blue contour lines were added onto the test strip for visual identification of the control and test line positions during the testing (Fig. 7A). The limit of detection for rCMY-34 was determined visually and reached 33 ng/mL. Due to limited sensitivity of the camera, a positive test line is not clearly visible in the picture when 33 ng/mL of rCMY-34 was tested (Fig. 7B). The ability of optimized LFIA-based test to detect CMY β-lactamases was examined with seven bacterial isolates producing CMY-2, CMY-4, CMY-6, CMY-16 and CMY-34 (Table 5). For analysis, a single colony of bacteria was mixed with extraction buffer and loaded onto the strip. The LFIA test was able to identify all tested CMY-positive isolates. No reactivity with CMY-negative bacteria, such as NDM-1 producing *A. chinensis* (CCUG74036T and CCUG74037) or β-lactamase-negative *E. coli* BL21, *K. aerogenes* (50796520) or *P. mirabilis* (50793946) was observed (Fig. 7C, Table 5).

MAbs 9D2 and 2E11 were further tested with TPX assay technique. The combination of MAbs showing the highest sensitivity was selected—MAb 9D2 coated microparticles were

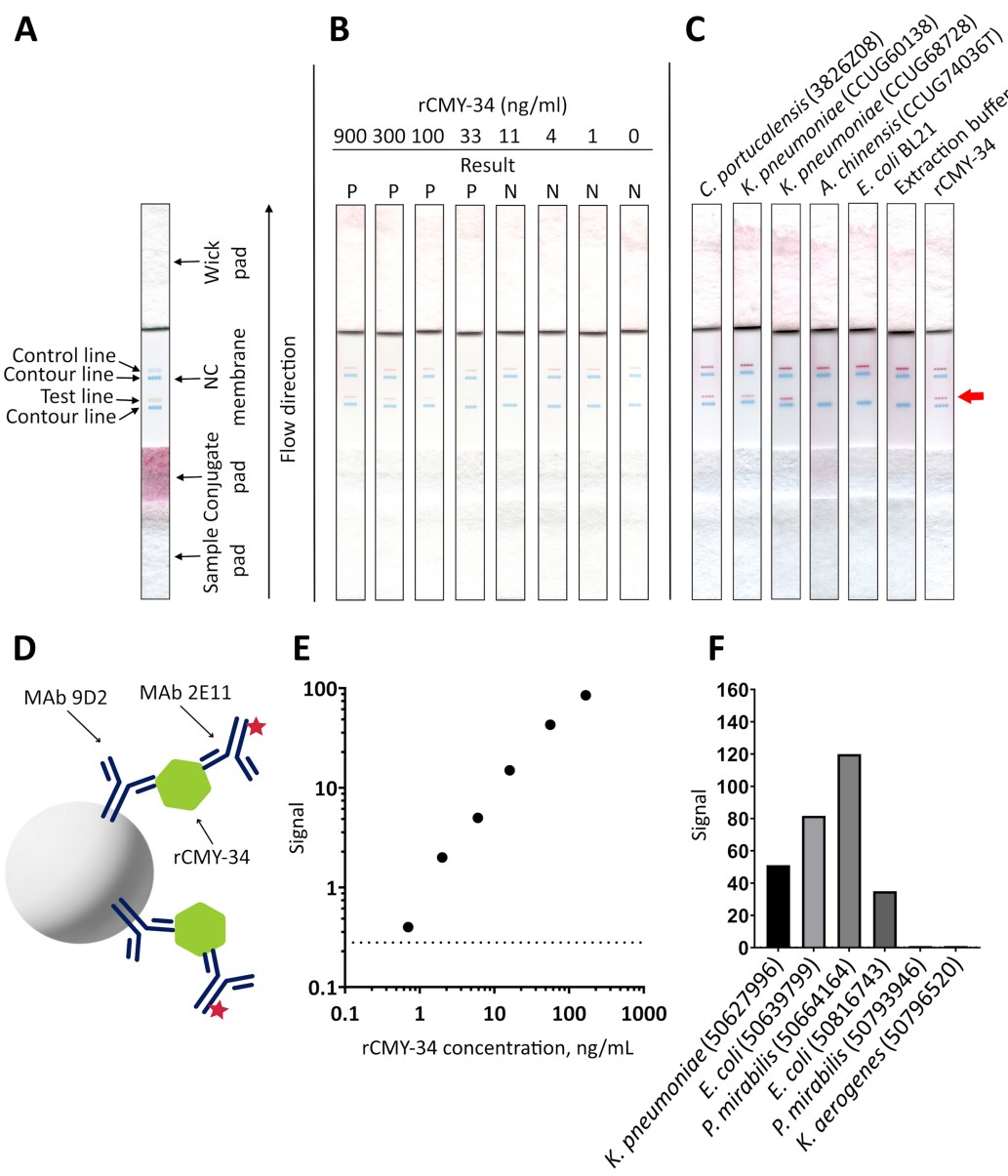

**Figure 7  Application of MAbs 9D2 and 2E11 in LFIA and their testing by TPX assay.** (A) Schematic representation of LFIA test strip. (B) Determination of limit of detection with rCMY-34 serial dilutions. "P" indicates a positive result, and "N" indicates a negative result when evaluated visually. (C) Testing of CMY-34 producing *C. portucalensis* and CMY-4 producing *K. pneumoniae* isolates. NDM-1-positive *A. chinensis*, *E. coli* BL21 strain and extraction buffer (used for sample preparation) were tested as negative controls. Red arrow indicates the position of positive test line. (D) Schematic representation of rCMY-34 detection with TPX assay. rCMY-34 is captured by MAb 9D2 coated microparticles and detected by fluorescently labelled MAb 2E11. (E) rCMY-34 dose-response curve (*n* = 3, dashed line represents a cut-off value 0.28). (F) Detection of bacterial isolates producing CMY β-lactamases (*n* = 3, cut-off value 0.28).                 

used for capturing the antigen, and fluorescently labelled MAb 2E11 was utilized for detection (Fig. 7D). The analytical sensitivity of the assay for rCMY-34 was 0.33 ng/mL, as defined by the value intercept of dose-response curve with three standard deviations

**Table 5 Bacterial isolates used for MAb characterization and evaluation of MAb-based immunoassays.**

| Species | Identification number | The presence of β-lactamases[a] | LFIA[c] | TPX[c] |
|---|---|---|---|---|
| *C. portucalensis* | 3826Z08 | **CMY-34** | + | NT |
| *K. pneumoniae* | CCUG60138 | NDM-1, **CMY-4**[b] | + | NT |
| *K. pneumoniae* | CCUG68728 | NDM-1, **CMY-4**[b] | + | NT |
| *A. chinensis* | CCUG74036T | NDM-1 | – | NT |
| *A. chinensis* | CCUG74037 | NDM-1 | – | NT |
| *K. pneumoniae* | 50627996 | **CMY-6**, NDM-1, OXA-1, CTX-M-15, SHV-11 | + | + |
| *E. coli* | 50639799 | **CMY-4**, OXA-1, VIM-29, CTX-M-15 | + | + |
| *P. mirabilis* | 50664164 | **CMY-16**, NDM-1, OXA-10 | + | + |
| *E. coli* | 50816743 | **CMY-2**, OXA-1, OXA-181, CTX-M-15, TEM-1 | + | + |
| *K. aerogenes* | 50796520 | Negative | – | – |
| *P. mirabilis* | 50793946 | Negative | – | – |

Notes:
[a] CMY β-lactamases are bolded.
[b] *bla*$_{CMY}$ was detected by PCR, and CMY allelic variant was identified by sequencing of PCR product.
[c] Positive result (+), negative result (–), not tested (NT).

(Fig. 7E). Positive results were obtained for all CMY producing isolates with high signal intensity (Fig. 7F, Table 5). No reactivity was observed towards the β-lactamase-negative *K. aerogenes* (50796520) and *P. mirabilis* (50793946) isolates.

# DISCUSSION

Widespread resistance to β-lactam antibiotics mediated by cephalosporinases, such as extended-spectrum (ESBL) or AmpC β-lactamases poses a potential risk to humans, livestock and wild animals worldwide (*Palmeira et al., 2021*). The cephalosporins are the second largest class of β-lactam antibiotics applied for the treatment of human and food-producing animal infections (*Chambers et al., 2015*). CMY-2-like β-lactamases are the most prevalent cephalosporinases globally (*European Committee on Antimicrobial Susceptibility Testing, 2017*). Therefore, easy-to-perform and accurate immunoassays detecting CMY-type β-lactamases producing antibiotic-resistant bacteria are epidemiologically relevant, leading to more effective treatment of infectious diseases and wider knowledge of CMY prevalence and transmission routes. CMY-34 is one of the barely investigated allelic variants of the CMY family (*Zhu, Xu & Xu, 2007*; *Hammerum et al., 2011*; *Damborg et al., 2012*). During this study, it was observed that CMY-34 exhibits a high sequence similarity to CMY-type β-lactamases, making this allelic variant a noticeable target for development of CMY-specific MAbs and their application in CMY-targeting AmpC detection assays.

In the present study, MAbs against CMY-34 β-lactamase were described for the first time. The use of recombinant CMY-34 as an immunogen allowed to generate a collection of IgG MAbs and select the highest affinity non-competing pair of antibodies produced by hybridoma clones 9D2 and 2E11. The properties of novel MAbs were characterized in detail by a wide range of immunoassays. Analysis of CMY producing bacterial isolates by

these methods revealed the ability of MAbs 9D2 and 2E11 to recognize CMY-34 and CMY-4 β-lactamases in *C. portucalensis* and *K. pneumoniae* isolates, respectively. Cross-reactivity testing with recombinant β-lactamases representing A, B, C and D classes demonstrated the specificity of MAb 2E11 to CMY enzymes. A weak cross-reactivity of MAb 9D2 with class C β-lactamase (also termed AmpC) PDC-195 was observed. Nevertheless, a larger collection of various AmpCs (ACC, FOX, LAT, *etc.*) should be tested for detailed evaluation of MAb cross-reactivity features.

Presumptive broad reactivity of the MAb 9D2 was demonstrated by epitope mapping with synthetic peptides and *E. coli* expressed CMY-34 fragments. Analysis of MAb 9D2 binding site (22–49 aa of CMY-34) revealed its high similarity across the members of CMY family, indicating its potential broad reactivity with a wide range of CMY-type enzymes. In contrast, the abovementioned mapping strategy did not allow identification of MAb 2E11 epitope within the CMY-34 protein. Supposedly, the recognition site of this MAb is disrupted by fragmentation of the antigen. To overcome this limitation, a putative epitope of MAb 2E11 within the CMY-34 molecule was predicted by computational modeling, using determined variable domain sequences of the MAb. The most reliable model of MAb 2E11-CMY-34 interaction revealed a conformational epitope, which is composed of three adjacent regions localized at 38–44, 299–311 and 362–379 aa of CMY-34. This putative epitope is conserved in most CMY β-lactamases, also suggesting a possible broad reactivity of this MAb with CMY-type enzymes.

In addition to optimal use of antibiotics in clinical settings, emerging transfer routes of antimicrobial resistance (AMR) between human and animal populations are recognized as a major concern. It is assumed that in humans, livestock and wildlife co-occurring mechanisms of AMR, including resistance to cephalosporines, are the same or closely related (*Palmeira et al., 2021*). The transmission route of resistance mechanisms between commensal bacteria of humans and animals is associated with the food chain (*Silbergeld, Graham & Price, 2008*; *Owaid & Al-Ouqaili, 2025*). Moreover, on the World Health Organization (WHO) priority pathogens list of antibiotic-resistant bacteria, *Enterobacterales*, such as *Klebsiella* spp., *Enterobacter* spp., *Citrobacter* spp., *Serratia* spp., *etc.* are distinguished as one of the most critical groups of pathogens possessing a great threat to the healthcare system (*World Health Organization, 2017*). The resistance of these bacteria to third-generation cephalosporins, which is conferred through the production of ESBL or AmpCs (plasmid or chromosomally transmitted), poses a significant threat to the treatment of common infections. This problem highlights the need for accurate and easy-to-perform assays for AMR testing and evaluation of prevalence.

Currently, several commercial LFIA tests targeting five major carbapenemases, belonging to class A, B and D β-lactamases, with 96–99% sensitivity and 100% specificity are available. For instance, NG Biotech (France) offers the NG-Test CARBA-5, Coris BioConcept (Belgium)—O.K.N.V.I. RESIST-5 test, and Era Biology (China)—K.N.I.V.O. K-Set test for detection and differentiation of NDM, IMP, VIM, OXA-48 and KPC β-lactamases (*Hong, Kang & Kim, 2021*; *Saito et al., 2022*; *Sadek et al., 2022*). For detection of ESBL, the NG-Test CTX-M MULTI (NG BiotechGuipry, France, ), which detects major

groups of CTX-M-type β-lactamases, has been commercialized (*Bianco et al., 2020*). Nevertheless, no such test has been reported or commercialized for detection of AmpC β-lactamases.

In this study, potential application of newly developed and comprehensively characterized MAbs 9D2 and 2E11 for detection of antibiotic-resistant bacterial isolates was demonstrated by performing quantitative sandwich ELISA, LFIA and TPX assay. The optimized immunoassays were able to detect all tested CMY-positive bacterial isolates (7 in total), producing CMY-2, CMY-4, CMY-6, CMY-16 and CMY-34 β-lactamases. Nevertheless, a more detailed examination of the assays with a wider range of CMY producers, CMY allelic variants and AmpC members should be conducted.

This study describes broadly reactive and comprehensively characterized MAbs that represent a promising novel tool for investigation and detection of CMY-type β-lactamases. These MAbs might be applied for studies of CMY prevalence in humans and wildlife and evaluation of AMR transfer pathways by applying them in rapid immunoassays or biosensors.

## CONCLUSIONS

Novel broadly reactive MAbs against CMY β-lactamases and MAb-based immunoassays are reported. These comprehensively characterized MAbs can be applied for immunodetection of CMY β-lactamases in bacterial isolates and may serve as a useful tool for assessing of CMY prevalence, as well as evaluating AMR transmission pathways.

## ACKNOWLEDGEMENTS

We are grateful to Anette M. Hammerum from Statens Serum Institute (Copenhagen, Denmark) for collaboration and sharing of CMY-34 producing *C. portucalensis* isolate. The authors would like to thank to colleagues form Institute of Biotechnology, Life Sciences Center, Vilnius University: Rasa Petraitytė-Burneikienė, Laima Čepulytė and Vytautas Rudokas for providing recombinant β-lactamases used for cross-reactivity testing and Agnė Rimkutė for MAb against SARS-CoV-2 spike protein used in IP as isotype control.

### Funding

This research was funded by the Agency for Science, Innovation and Technology and the Research Council of Lithuania, grant no. 01.2.2-MITA-K-702-05-0003, "Novel affinity binders for immunodetection of antimicrobial resistance", and Research Council of Lithuania, grant no. S-MIP-23-44, "Deep learning-based methods for annotating protein interactions". There was no additional external funding received for this study. The funders had no role in study design, data collection and analysis, decision to publish, or preparation of the manuscript.

## Grant Disclosures

The following grant information was disclosed by the authors:

Agency for Science, Innovation and Technology and the Research Council of Lithuania: 01.2.2-MITA-K-702-05-0003.

Research Council of Lithuania: S-MIP-23-44.

## Competing Interests

Julie Nuttens is employed by ArcDia International Oy Ltd. The company specializes in the development and manufacturing of rapid diagnostic tests for rapid diagnostics of acute infections.

## Author Contributions

- Karolina Bielskė conceived and designed the experiments, performed the experiments, analyzed the data, prepared figures and/or tables, authored or reviewed drafts of the article, and approved the final draft.
- Martynas Simanavičius performed the experiments, analyzed the data, authored or reviewed drafts of the article, and approved the final draft.
- Julie Nuttens performed the experiments, analyzed the data, authored or reviewed drafts of the article, and approved the final draft.
- Julija Armalytė performed the experiments, authored or reviewed drafts of the article, and approved the final draft.
- Justas Dapkūnas performed the experiments, analyzed the data, prepared figures and/or tables, authored or reviewed drafts of the article, and approved the final draft.
- Lukas Valančauskas performed the experiments, analyzed the data, authored or reviewed drafts of the article, and approved the final draft.
- Aurelija Žvirblienė conceived and designed the experiments, authored or reviewed drafts of the article, and approved the final draft.

## Animal Ethics

The following information was supplied relating to ethical approvals (*i.e.*, approving body and any reference numbers):

Mice used for hybridoma technology were obtained from Vilnius University, Life Sciences Center, Institute of Biochemistry (Vilnius, Lithuania), which has State Food and Veterinary Agency (Vilnius, Lithuania) permission to breed and use experimental animals for scientific purposes (vet. approval no. LT 59–13–001, LT 60–13–001, LT 61–13–004). Ethical approval to use BALB/c mice for experiments was granted by State Food and Veterinary Agency (Vilnius, Lithuania), permission no. G2–117, issued 11 June 2019.

## DNA Deposition

The following information was supplied regarding the deposition of DNA sequences:

The determined VH and VL sequences of the MAbs 9D2 and 2E11 are available at the ABCD Database: ABCD_BD767 and ABCD_BD768.

https://web.expasy.org/abcd/ABCD_BD767.
https://web.expasy.org/abcd/ABCD_BD768.

## Data Availability

The raw datasets are available at MIDAS: Development of MAbs against CMY β-lactamases; http://dx.doi.org/10.18279/MIDAS.258798.

The structure models of MAb-CMY-34 interaction are available at ModelArchive:

- AlphaFold2 model of MAb 2E11-CMY-34 interaction: ma-u2d16; https://doi.org/10.5452/ma-u2d16.

- AlphaFold3 model of MAb 2E11-CMY-34 interaction: ma-80uzo; https://doi.org/10.5452/ma-80uzo.

- AlphaFold3 model of MAb 9D2-CMY-34 interaction: ma-22lwx; https://doi.org/10.5452/ma-22lwx.

## Supplemental Information

Supplemental information for this article can be found online at http://dx.doi.org/10.7717/peerj.20036#supplemental-information.

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
