# Peer review of "Novel monoclonal antibodies for immunodetection of AmpC β-lactamases"

_PeerJ, doi:10.7717/peerj.20036_

## Round 0.1 · original submission · Minor Revisions

·

Basic reporting

Dear Authors,
You are doing excellent work, and this manuscript reflects great effort and thoughtful research. The study is promising and contributes meaningfully to the field. However, there are a few important points that require further clarification and revision to enhance the overall quality and impact of the manuscript.
Lines 45–46: I recommend including recent research on antibiotic resistance mechanisms to strengthen the manuscript's discussion about the growing global challenge of bacterial resistance: Rawaa A. Hussein, Shaymaa H. AL-Kubaisy, Mushtak T.S. Al-Ouqaili. The influence of efflux pump, outer membrane permeability and β-lactamase production on the resistance profile of multidrug, extensively and pandrug-resistant Klebsiella pneumoniae. Journal of Infection and Public Health, Volume 17, Issue 11, 2024. https://doi.org/10.1016/j.jiph.2024.102544.
Lines 49–51: The classification of AmpCs as class C β-lactamases is accurate and appropriately referenced. The sentence regarding resistance to cephalosporins and β-lactam/β-lactamase inhibitor combinations is informative, but it contains a recurring encoding issue that needs to be resolved.
Lines 50–58: To enhance the introduction section, add the following reference: AL-KUBAISY SH, HUSSEIN, RA, AL-OUQAILI, MTS. (2020). Molecular Screening of Ambler class C and extended spectrum β-lactamases in multidrug-resistant Pseudomonas aeruginosa and selected species of Enterobacteriaceae. International Journal of Pharmaceutical Research | Jul - Sep 2020 | Vol 12 | Issue 3
Lines 59–65: The use of the Beta-Lactamase DataBase (BLDB) and the mention of over 180 black variants (with the 2025 update) strengthens the argument. Additionally, including a sentence on the clinical significance of these variants would emphasise their relevance.
Lines 66–72: The literature strongly supports the history of CMY-34 detection. However, this study could benefit from a clearer transition explaining why CMY-34 is the focus and how it structurally or functionally differs from other CMY variants.
Lines 73–81: To enhance the introduction section, add the following reference: Al-Ouqaili, MTS, Al-Taei, SA, Al-Najja,r A. Molecular Detection of Medically Important Carbapenemases Genes Expressed by Metallo-β-lactamase Producer Isolates of Pseudomonas aeruginosa and Klebsiella pneumoniae. Asian Journal of Pharmaceutics • Jul -Sep 2018 (Suppl ) • 12 (3) | S991.
Lines 94–99: The distinction between commercial and molecular methods is clear. However, the mention of boronic acid derivatives inhibiting some class A β-lactamases should be slightly expanded to explain the diagnostic challenge this poses.
Lines 109–115: The explanation of TPX technology is both interesting and novel. However, it would benefit from further clarification on how TPX compares to existing rapid tests regarding sensitivity, cost, and field usability.
Line 130: It is commendable that the synthetic gene source and GenBank accession number are provided. Still, a brief description of the gene sequence features (e.g., native vs. codon-optimised) would be informative.
Line 133: While the construct was verified by sequencing, details about the sequencing method (e.g., Sanger sequencing) and coverage should be included to enhance the reliability of the construct validation.
Line 157: While the sample size was determined based on standard practices, this reasoning remains unclear. A concise justification or reference to guidelines (e.g., FELASA recommendations) for using three mice would enhance methodological transparency.
Lines 158–159: Acknowledging the absence of randomisation, control groups, or blinding is appreciated. However, it is important to provide a brief justification for why these methods were not applicable or necessary in this context, as this will help alleviate concerns about potential bias.
Lines 289–290:The authors cite Stravinskiene et al. (2020) for the sequencing procedure. While referencing is acceptable, providing a summary of key steps (e.g., RNA extraction, cdna synthesis method, and polymerase used) would enhance clarity and reproducibility.
Lines 422–424: The production of 14 Igg-class MAbs is well stated. However, clarity would improve by briefly noting how many distinct clones were screened and whether isotyping was performed for all.
Lines 454–457: Strong demonstration of MAb specificity and cross-reactivity with both CMY-34 and CMY-4. However, clarify whether signal intensities differed significantly across strains (e.g., relative protein expression).
Lines 592–599: Two-photon excitation (TPX) provides high sensitivity and represents an innovative approach for rapid diagnostics. Clearly stating that MAb 9d2 was immobilised and MAb 2e11 was fluorescently labelled enhances the understanding of assay architecture.
The LOD of 0.33 ngmlL is excellent—likely the most sensitive among the tested methods. Ensure the definition of LOD (“three SD from intercept”) aligns with standard analytical guidelines.
To enhance the discussion section, add the following reference: Al-Ouqaili, MTS., Jal'oot, AS., Badawy, AS. Identification of an Oprd and bla(IMP) Gene-mediated Carbapenem Resistance in Acinetobacter baumannii and Pseudomonas aeruginosa among Patients with Wound Infections in Iraq. Asian Journal of Pharmaceutics, volume 12, Issue 3, 2019, Page S959-S965.
Line 611: CMY-34 was selected for immunogen development due to its unique properties, which can potentially provide better coverage and effectiveness compared to more common variants like CMY-2.
To enhance the discussion section, add the following reference: Hekmat A. Owaid, Mushtak T.S. Al-Ouqaili, Molecular characterisation and genome sequencing of selected highly resistant clinical isolates of Pseudomonas aeruginosa and its association with the clustered regularly interspaced palindromic repeat/Cas system, Heliyon, Volume 11, Issue 1, 2025, e41670, ISSN 2405-8440, https://doi.org/10.1016/j.heliyon.2025.e41670.
Line 674: Conclusion should be objective with further perspective,o r should add at least a few sentences about future study/future perspective of it

Experimental design

I think the Original primary research is within the Aims and Scope of the journal.The other assigned items are ok

Validity of the findings

Everything is Ok regarding the assigned items

Reviewer 2 ·

Basic reporting

The study aims to identify novel monoclonal antibodies for the immunodetection of CMY β-lactamases. The author did an excellent job, and the overall findings will positively impact the relevant research.
The article is written in professional English and is easy to follow and comprehend.
The introduction properly addressed the reason for conducting this research with adequate background information. This research is timely and was performed to address the concerns about antibiotic resistance.

Experimental design

The study is structurally well formulated, and the author conducted thorough experiments to back the findings.

Validity of the findings

The authors provided all the necessary data with proper control and statistical analysis to draw the conclusion independently.

Additional comments

The article can be accepted with the following minor corrections-
45, please rewrite the sentence.
84-115, unless the author really feels compelled to include all those details, please try to reduce the writing and just point out the most important techniques, as detailing all those testing techniques diverts the reader's focus and overwhelms them with information.
125-128, this section seems repetition of the same information that is written later in a detailed form. I would suggest deleting this point. And incorporate the E. coli BL21 strain usage in the next assay where it was used.
188-190, Mice would have been… as no such occurrence occurred, and it is part of approved animal protocol, this sentence can be deleted.
432, there is a typo in IgG2a, Kd 0,10 nM.

Reviewer 3 ·

Basic reporting

The manuscript is well written in professional English and sufficient references and background are provided. Table and figure captions generally provide enough information for readers to understand.

Experimental design

In this study, the authors reported the production and characterization of two novel monoclonal antibodies, 9D2 and 2E11, and demonstrated their applications for detection of antibiotic-resistant bacterial isolates producing AmpC beta-lactamases. In general, most claims are supported by the experimental design.

Validity of the findings

1. It seems to be a typo of the marker labels in Figure 1a, should be 40 instead of 45?
2. In Figure 1c, some smaller protein bands can be observed in the samples C. portucalensis and rCMY-34. Are they degraded proteins of CMY-34 or something else? It would be nice to specify what those bands are from.
3. In Figure 1b, the authors claimed and showed in WB that CMY-4 produced from K. pneumoniae can be bound by 9D2 and 2E11. It would be nice to provide quantitative analysis on how well the binding is.
4. In the "Computational prediction of MAb 2E11 epitope" section, 6000 AlphaFold models were reported to be generated and 20 of them were picked for downstream analysis. It would be ideal to provide a confidence score distribution figure for 6000 models and an average RMSD of 19 similar models. In addition to the computational prediction of MAb 2E11, I am curious if the prediction of MAb 9D2 has been done. The computational prediction of MAb 2E11 can be strengthened if the prediction of MAb 9D2 matches the experimental results of epitope mapping. For the computational prediction, have you tried AlphaFold3? I guess AlphaFold3 might work better in your case.
5. In lines 665–667, the authors claimed that the optimized immunoassays were able to detect all tested CMY-positive bacterial isolates producing CMY-2, CMY-4, CMY-6, CMY-16 and CMY-34 beta-lactamases. However, I can only find the related experimental results in Table 5 in a qualitative manner (+ and –). I think more results should be shown to support this important claim.

Additional comments

no comments

·

Basic reporting

No comment

Experimental design

No comment

Validity of the findings

No comment

Additional comments

General comments
The study highlights the development of broadly reactive MAbs against CMY-type A Beta-lactamases, which can be utilized in rapid immunoassays for detecting antibiotic-resistant bacteria. These MAbs represent a promising tool for monitoring CMY prevalence and AMR transfer pathways. The study focused on the reactivity of monoclonal antibodies (MAbs) against CMY-34 β-lactamases, demonstrating their effectiveness in various immunoassays. ​ The findings highlight the potential for these MAbs in detecting antimicrobial resistance.

1. MAbs 9D2 and 2E11 showed reactivity against CMY-34 and CMY-4 in Western Blot (WB) and ELISA tests. ​
2. The study included cross-reactivity analysis with other β-lactamases.
MAb 9D2 epitope mapping identified specific amino acid sequences critical for recognition.
The study design and experimentation were robust and suffice for this kind of work.

Specific comments:
Line 201: Mention PBST in full at its first mention
Line 208: Mention OD in full at its first mention
Line 215: Mention RT in full at its first mention
Line 224: Introduce citation showing reference to ELISA protocol
Line 224: Don’t begin a sentence with initials
Line 228: Add a text citation
Line 281:Mention IPTG in full at its first mention

---

## Round 0.2 · accepted · Accept

The authors have addressed all of the reviewers' comments.

·

Basic reporting

-

Experimental design

-

Validity of the findings

-

Reviewer 2 ·

Basic reporting

The author has addressed all the concerns, and the article can be published as is.

Experimental design

-

Validity of the findings

-

Reviewer 3 ·

Basic reporting

The manuscript is well written in professional English, and sufficient references and background are provided. Table and figure captions generally provide enough information for readers to understand.

Experimental design

In this study, the authors reported the production and characterization of two novel monoclonal antibodies, 9D2 and 2E11, and demonstrated their applications for the detection of antibiotic-resistant bacterial isolates producing AmpC beta-lactamases. In general, the major claims are supported by the experimental design.

Validity of the findings

It is exciting to see that AlphaFold3 worked better on MAb 9D2 and generated more reliable models that support the experimentally determined epitope site.

·

Basic reporting

The language used throughout the abstract, introduction, and methods sections appears clear, unambiguous, and technically correct. It conforms to professional standards of courtesy and expression. The introduction effectively provides sufficient background and context regarding the increasing prevalence of antibiotic-resistant bacteria, specifically AmpC β-lactamases, and the clinical importance of their detection. Relevant prior literature is extensively referenced, including established guidelines and previous studies on AmpC detection methods.

Experimental design

This is clearly original primary research focused on developing novel monoclonal antibodies and immunoassays for AmpC β-lactamases, specifically CMY-type. This topic is highly relevant and fits within the typical aims and scope of this journal. The paper clearly identifies the knowledge gap, stating that "no such assay or monoclonal antibodies (MAbs) capable of detecting other CMY β-lactamases have been described yet", and that "rapid tests such as TPX assay or lateral flow immunoassay (LFIA) for immunodetection of CMY-type β-lactamases in bacterial samples are not available yet". The study directly addresses this gap by describing broadly reactive MAbs and their application in sandwich ELISA, LFIA, and TPX assays. The experiments appear technically rigorous and comprehensive. Detailed procedures, reagents, and equipment with catalog numbers are provided.

The paper demonstrates a strong commitment to ethical standards in animal experimentation. It explicitly states adherence to ARRIVE and FELASA guidelines, European and Lithuanian legislations, and provides specific approval numbers for animal use. Details about animal care, monitoring, and sacrifice are provided, indicating attention to animal welfare.

Validity of the findings

The conclusion concisely states that the novel MAbs raised against CMY-34 recognize common epitopes of CMY β-lactamases and can be applied for immunodetection in bacterial isolates. This directly links back to the stated aim and is well-supported by the results, which indicate that the MAb-based immunoassays detected all analyzed CMY-positive isolates. The conclusions appear to be limited to those supported by the presented results. This paper presents original research. The description of Kd determination from titration curves and analysis of MAb epitopes suggests statistically sound approaches for antibody characterization.

Additional comments

The writing is professional, clear, and unambiguous, making the complex scientific content accessible. The methods used for antibody generation, characterization, and immunoassay development are described in sufficient detail to ensure reproducibility. Overall, the paper demonstrates a high standard in its reporting and experimental design, contributing valuable original research to the field of infectious disease diagnostics.